# The Geometry of Reasoning: Self-Evaluation via Layerwise Trajectory Evolution

**Jinhe Bi** [1 2 3 4]  **Danqi Yan** [2]  **Yifan Wang** [2]  **Wenke Huang** [5]  **Haokun Chen** [2]  **Guancheng Wan** [6]  **Mang Ye** [6]
**Xun Xiao** [1]  **Hinrich Schuetze** [2 3]  **Volker Tresp** [2 3]  **Yunpu Ma** [2 3 4]

## Abstract

Large Reasoning Models (LRMs) enhance performance by generating explicit Chain-of-Thought (CoT) trajectories, yet enabling them to self-evaluate correctness without external supervision remains a critical challenge. Existing methods often rely on ground-truth labels or shallow output probabilities, neglecting the layerwise evolution of the reasoning trajectory. In this work, we introduce `GeoR` (Geometry of Reasoning), a white-box self-evaluation framework based on layerwise trajectory evolution. `GeoR` decomposes reasoning fidelity into two complementary dimensions: (1) Geometric Evolution, which synthesizes the first- and second-order evolution of layerwise hidden-state trajectories to quantify geometric progress in reasoning; and (2) Difficulty-Aware Calibration, which utilizes cross-entropy of reasoning progress to normalize the Geometric Evolution against intrinsic query uncertainty. By jointly modeling these factors, `GeoR` effectively distinguishes the coherent evolution of correct reasoning from the chaotic trajectories of errors. Extensive experiments across eight LRMs and seven benchmarks demonstrate that `GeoR` consistently outperforms state-of-the-art baselines in AUROC, AUPR, and FPR@95. The code is available at [this repository](#).

## 1. Introduction

Large Reasoning Models (LRMs) (DeepSeek-AI, 2025; OpenAI, 2024; Team, 2025c; Bi et al., 2026b) have achieved notable progress on complex reasoning tasks (Chen et al., 2023; Cobbe et al., 2021; Hendrycks et al., 2021; Shi et al., 2023; Wang et al., 2024a; Talmor et al., 2019; Tan et al.,

[1]Huawei Heisenberg Research Center [2]LMU Munich [3]Munich Center for Machine Learning [4]MemAgents Lab [5]College of Computing and Data Science, Nanyang Technological University, Singapore [6]Wuhan University. Correspondence to: Jinhe Bi <bijinhe@outlook.com>, Yunpu Ma <cognitive.yunpu@gmail.com>.

*Proceedings of the $43^{rd}$ International Conference on Machine Learning*, Seoul, South Korea. PMLR 306, 2026. Copyright 2026 by the author(s).

2026; Yuan et al., 2025b; 2026) by generating answers through structured and deliberate reasoning trajectories. However, while such trajectories sometime are capable of yielding correct answers, they may also culminate in incorrect or even harmful outputs, raising concerns about the reliability of model predictions. To address this issue, various approaches have been proposed to evaluate LRM reliability. They can be broadly grouped into two categories, which are briefly discussed as follows.

The first relies on external assessment, leveraging human annotations, linear probing, or auxiliary judge models (Wang & Xu, 2025; Li et al., 2023; He et al., 2025). While effective in controlled, supervised settings, these external assessment methods lack generality and are impractical at deployment: they depend on external resources that may be unavailable and degrade significantly under distribution shift between training and real-world queries.

The second focuses on *self-evaluation* that estimates correctness based solely on the model's own internal signals, in our case study illustrated in Figure 1. This can be further divided into white-box and black-box approaches: white-box methods quantify internal uncertainty measures—such as softmax confidence, temperature scaling, or hidden-state entropy—without external supervision (Wang et al., 2025a); and black-box methods directly prompt the model to output confidence scores over its responses (Sun et al., 2025). However, the existing self-evaluation methods still face critical challenges. Black-box approaches are systematically over-confident and offer limited interpretability (Sun et al., 2025; Wang et al., 2025a). White-box approaches (Shih et al., 2023; Huang et al., 2025b; Si et al., 2023) often collapse the entire reasoning process into a single-step output distribution, leaving the rich trajectory $z_{0:T}$ underutilized. This deficiency in the prior art brings up an open question:

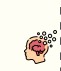 *Can we enable LRMs to better self-evaluate their outputs by leveraging their own reasoning trajectories?*

**The Present Work:** Our solution, `GeoR`, addresses this question by exploiting the rich dynamics of layerwise tra-

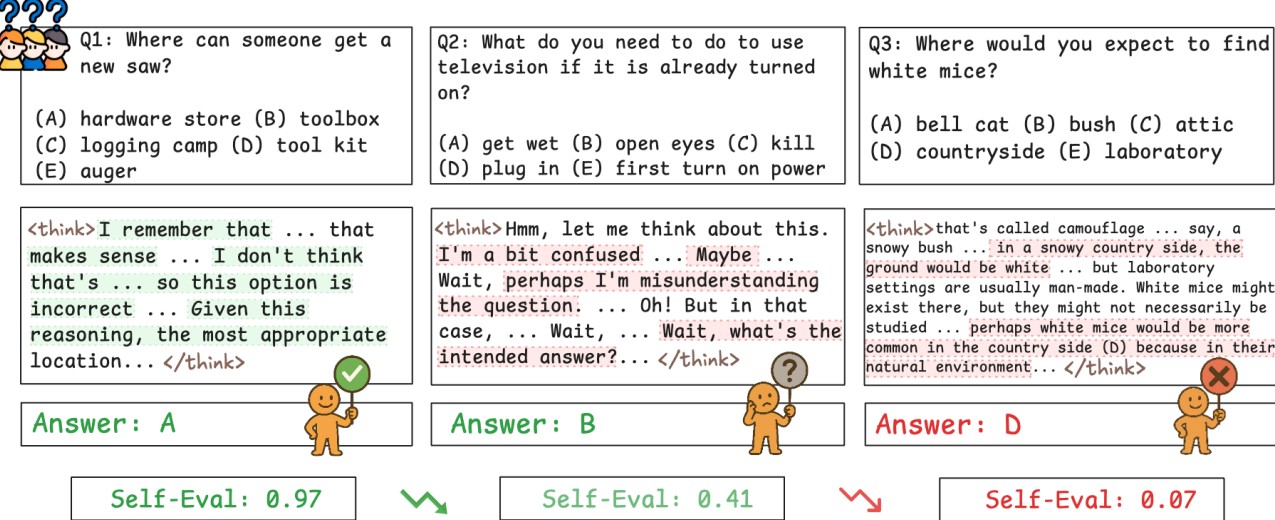

*Figure 1.* **Illustration of Self-Evaluation.** Self-evaluation aims to assess the correctness of model outputs solely based on the generated reasoning trajectory. As illustrated, the goal is to assign a reliability score that correlates with reasoning quality: correct and coherent trajectories (Q1) should receive high confidence scores (e.g., 0.97), whereas trajectories exhibiting confusion (Q2) or reasoning errors (Q3) should be flagged with low scores (e.g., 0.41, 0.07). This enables the model to internally verify its answers without external supervision.

jectory evolution—a dimension largely overlooked by existing studies. However, establishing a relationship between layerwise hidden-state trajectories in LRMs and final output correctness without supervision presents a fundamental challenge. Prevailing self-evaluation methods are typically "state-blind": they rely exclusively on surface-level output token metrics, thereby neglecting the deep, geometric evolution of the reasoning process internal to LRMs.

To bridge this gap, we leverage the fundamental mechanism of LRMs, where stacked transformer layers iteratively transform hidden states to drive trajectory evolution. We posit that the quality of this evolution—whether reasoning progresses effectively or stagnates—is naturally encoded in the layer-to-layer hidden-state updates. Consequently, to quantitatively profile this process, we utilize the first-order and second-order variations of hidden states to characterize their *Geometric Evolution*.

As visualized in Figure 2, we analyze this Geometric Evolution across benchmarks of varying complexity. We observe a consistent pattern: simpler tasks (e.g., GSM8K) exhibit systematically higher Geometric Evolution, reflecting decisive progress in the hidden-state space. In contrast, complex tasks (e.g., MMLU) often result in diminished evolution magnitudes. Crucially, this geometric distinction extends to identifying correctness within a single task: correct reasoning trajectories consistently manifest higher Geometric Evolution, while incorrect outputs tend to exhibit stagnation or unstable evolution. These findings confirm that the evolution of layerwise hidden-state trajectories provides a principled, white-box signal for self-evaluating reasoning fidelity (see Section 2.2.1 for details). Furthermore, we

introduce the *Difficulty-Aware Calibration*, which utilizes cross-entropy to normalize Geometric Evolution against intrinsic query uncertainty (see Section 2.2.2 for details). We term this unified framework `GeoR` (Geometry of Reasoning). Detailed formulations and empirical demonstrations are presented in the following sections.

We conduct comprehensive evaluations to assess the effectiveness and generality of `GeoR`. Across eight open-source LRMs spanning 1.5B to 70B parameters (covering multiple LLMs families and vision–language architectures such as Qwen-VL), diverse training paradigms (SFT, distillation, and RL/GRPO), and seven benchmarks in mathematics, commonsense reasoning, multilingual understanding, multimodal reasoning, theorem proving, general knowledge, and spatial understanding, `GeoR` consistently delivers strong and stable performance. This robustness across model scales, architectures, training recipes, and task families indicates that `GeoR` captures intrinsic reasoning reliability rather than incidental geometric artifacts, and it remains stable even under open-world, mixed-distribution settings where existing baselines noticeably degrade.

Our key contributions can be summarized as follows:

❶ **Geometric Evolution and Difficulty Calibration.** We identify that reasoning fidelity is governed by two complementary dimensions: Geometric Evolution, which synthesizes layerwise hidden-state dynamics to capture the quality of reasoning progress, and Difficulty-Aware Calibration, which leverages entropy to normalize this evolution against intrinsic query uncertainty. Together, these components provide a principled basis for distinguishing coherent reasoning from incorrect trajectories.

❷ **The `GeoR` Framework.** We propose `GeoR` (Geometry of Reasoning), a novel white-box self-evaluation framework that integrates Geometric Evolution with Difficulty-Aware Calibration. Unlike existing methods that rely solely on shallow output probabilities, `GeoR` exploits the deep geometric evolution of the reasoning process, offering a more robust and interpretable measure of output correctness without requiring external supervision.

❸ **Extensive Empirical Validation.** Comprehensive experiments across eight LRMs (ranging from 1.5B to 70B parameters) and seven diverse benchmarks demonstrate that `GeoR` is **(I)** *high-performing*, consistently surpassing state-of-the-art baselines across varying model scales, architectures, and task domains, and **(II)** *resource-friendly*, maintaining lower computational cost than mainstream self-evaluation methods.

## 2. Self-Evaluation via Layerwise Trajectory Evolution

### 2.1. Formulation of LRM CoT Reasoning

LRMs unfold reasoning as a structured and deliberative process, where the model generates not only a conclusive answer but also a reasoning trajectory together as the outcome of inference, as the examples shown in Figure 1. Upon receiving a query task, an LRM will extract and encode the information contained in the question into a unified representation to facilitate the subsequent answering with thinking processes. Following the query embedding processing, the model starts its reasoning phase, where it generates intermediate reasoning tokens between designated special tokens such as `<think>` and `</think>`, which will be eventually translated into textual information as the reasoning trajectory.

For this specific reasoning phase, we assume the total sequence length of generated CoT reasoning tokens is $K$, wherein the $k$-th generated reasoning token is denoted as $q_k$. We further formalize the layer-wise inference process in the transformer for generating $q_k$ as follows. At each transformer layer, the hidden state of each token is updated based on the model parameters and the accumulated processing results from the previous layers. Formally, let $q_k^i \in \mathbb{R}^d$ denote the hidden state of the $k$-th token modified after the $i$-th layer computation, where $d$ is the hidden dimension. The state transition from $q_k^{i-1}$ to $q_k^i$ can be represented as a transformation function below:

$$q_k^i = F_i(q_k^{i-1}, W^i), \ \forall i = 1, 2, \cdots, L \,, \quad (1)$$

where $W^i$ represents the synthesized effect of the model parameters and attention mechanisms applied to the $q_k^{i-1}$; and $L$ is the total number of transformer layers of an LRM.

Once the final reasoning token $q_K^L$ is produced, the reasoning phase concludes and the model transitions to generating the final answer conditioned on the trajectory. In this work, we propose to profile the entire reasoning trajectory and derive `GeoR` from its Geometric Evolution, enabling reliable self-evaluation of model outputs. We first apply mean pooling over the collected hidden states that are modified by the same transformer layer $i$ across all reasoning tokens. Specifically, we average all $K$ reasoning tokens generated by layer $i$, i.e., $\bar{q}^i = \frac{1}{K} \sum_{k=1}^{K} q_k^i$, where $\bar{q}^i$ serves as a compact semantic summary of the reasoning trajectory at the $i$-th layer, aggregating the representational states of all intermediate reasoning steps excluding the final answer tokens.

The main purpose of doing the mean pooling above is explained as follows. First of all, since these tokens are generated within the dedicated reasoning phase (i.e., between `<think>` and `</think>`), their averaged representation preserves the core semantics relevant to self-evaluation while suppressing token-level noise (empirically validated in Section 3.4, Table 2). Moreover, mean pooling operation reduces variance and highlights the global progression of reasoning across layers, which is critical for modeling the layer-wise Geometric Evolution. Note that this simplification strategy has been widely adopted in prior work for semantic representation modeling (Gao et al., 2021; Reimers & Gurevych, 2019; Bi et al., 2026a).

### 2.2. `GeoR`: Geometric Evolution and Difficulty Calibration

#### 2.2.1. GEOMETRIC EVOLUTION

Recent studies suggest that the hidden states of LRMs encode rich internal signals that strongly correlate with response correctness (Azaria & Mitchell, 2023; Liu et al., 2023). Unlike prior approaches that require correctness labels to train probing classifiers (Su et al., 2024; Burns et al., 2023), we analyze the intrinsic *Geometric Evolution* of the reasoning trajectory.

A critical design choice in our framework is modeling this evolution along the *layer dimension* ($L$) rather than the temporal token sequences ($K$). While token generation unfolds in time, we posit that the true "reasoning progression" occurs along the network depth, where intermediate representations are iteratively refined and stabilized. This layer-wise evolution acts as a proxy for the model's internal "depth of thought," offering a stable signal of reasoning quality that is robust to the high variance and lexical noise often found in surface-level token dynamics (empirically validated in Section 3.4, Table 2).

We quantify this evolution by decomposing the hidden-state updates into two geometric components: first-order progress ($D_1$) and second-order adaptability ($D_2$). Collecting the hidden representations into a $(K, L, d)$ tensor, we first define the layerwise hidden-state variations as:

$$\Delta \bar{q}^i = \bar{q}^i - \bar{q}^{i-1} \,, \qquad \Delta^2 \bar{q}^i = \Delta \bar{q}^i - \Delta \bar{q}^{i-1} \,, \quad (2)$$

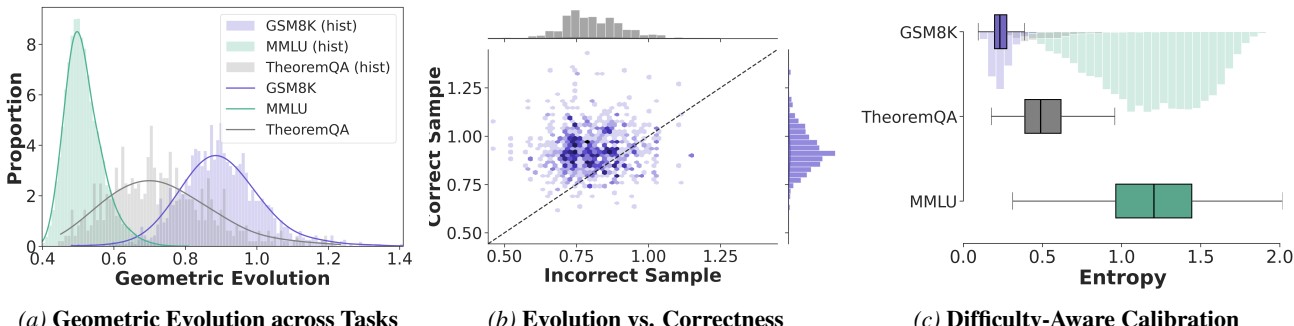

*(a)* **Geometric Evolution across Tasks**    *(b)* **Evolution vs. Correctness**    *(c)* **Difficulty-Aware Calibration**

*Figure 2.* **Analysis of Geometric Evolution and Difficulty.** (a) *Geometric Evolution* varies with task complexity: simpler tasks (e.g., GSM8K) exhibit stronger reasoning progress than complex ones (e.g., MMLU). (b) Correct trajectories consistently manifest higher Geometric Evolution scores than incorrect ones, validating it as a signal for correctness. (c) Cross-entropy aligns with task difficulty, supporting its use for *Difficulty-Aware Calibration* to normalize evolutionary progress.

where $\bar{q}^i$ is the layer-averaged hidden state at layer $i$. Based on these vector variations, we define the scalar geometric components as the accumulated layerwise magnitudes: $D_1 = \sum \|\Delta \bar{q}^i\|$ and $D_2 = \sum \|\Delta^2 \bar{q}^i\|$. Here, $D_1$ measures the total effective progress, while $D_2$ captures the aggregate curvature of the trajectory, reflecting adaptive adjustments. It is worth noting that while these geometric quantities are derived from Euclidean norms, our final framework naturally explicitly normalizes them by the global trajectory displacement to ensure they are dimensionless and strictly comparable with probabilistic signals (detailed in Section 2.2.2).

As visualized in Figure 2a, we analyze this Geometric Evolution (quantified as the summation of $D_1$ and $D_2$) across benchmarks of varying complexity. We observe a consistent pattern: simpler tasks (e.g., GSM8K) exhibit systematically higher Geometric Evolution, reflecting decisive progress in the hidden-state space. In contrast, complex tasks (e.g., MMLU) often result in diminished evolution magnitudes. Crucially, this geometric distinction extends to identifying correctness within a single task: correct reasoning trajectories consistently manifest higher Geometric Evolution, while incorrect outputs tend to exhibit stagnation or unstable dynamics as shown in Figure 2b. These findings confirm that the evolution of layerwise hidden-state trajectories provides a principled, white-box signal for self-evaluating reasoning fidelity.

### 2.2.2. DIFFICULTY-AWARE CALIBRATION

To further enhance self-evaluation, we introduce *Difficulty-Aware Calibration* to account for intrinsic query uncertainty. We quantify this using the average negative log-likelihood (NLL) of the reasoning trajectory actually generated by the model. Specifically, let $q_1, \ldots, q_K$ be the sequence of reasoning tokens produced by the model. We define the difficulty $\mathcal{H}$ as the average uncertainty over the full sequence:

$$\mathcal{H} = \frac{1}{K} \sum_{k=1}^{K} -\log p(q_k | q_{<k}, \mathbf{x}), \quad (3)$$

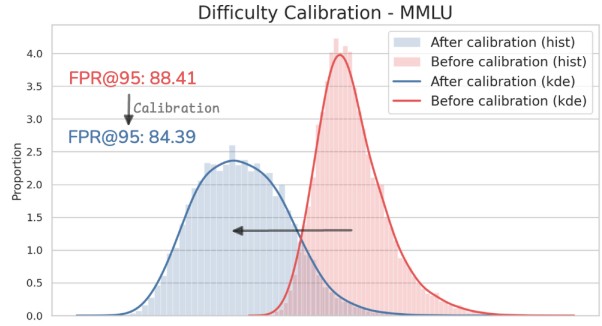

*Figure 3.* **Effect of Difficulty-Aware Calibration on MMLU.** Incorporating calibration ($\mathcal{H}$) smooths the score distribution and separates reliable trajectories from high-uncertainty ones, significantly reducing FPR@95.

where $p(q_k | \ldots)$ denotes the probability of the token $q_k$ actually selected by the model. This measure captures the intrinsic uncertainty of the reasoning process: higher $\mathcal{H}$ implies the model 'struggled' with the query, facing greater uncertainty at each step.

Integrating $\mathcal{H}$ is crucial because geometric dynamics alone can be ambiguous. As observed in Figure 2a and Figure 2c, trajectories with similar geometric magnitude may stem from vastly different confidence levels. A high-entropy trajectory suggests chaotic exploration rather than decisive reasoning, even if the geometric step sizes are large. By penalizing such uncertainty, we define the calibrated `GeoR` score as:

$$\texttt{GeoR} = \underbrace{\frac{D_1 + D_2}{\|\bar{q}^L - \bar{q}^0\|}}_{\text{Normalized Geometric Drive}} - \underbrace{\mathcal{H}}_{\text{Calibration}}$$

$$= \frac{\sum_{i=2}^{L} \|\Delta \bar{q}^i\| + \sum_{i=3}^{L} \|\Delta^2 \bar{q}^i\|}{\|\bar{q}^L - \bar{q}^0\|} \quad (4)$$

$$- \frac{1}{K} \sum_{k=1}^{K} -\log p(q_k | q_{<k}, \mathbf{x}).$$

Here, we explicitly normalize the geometric path length

$(D_1 + D_2)$ by the total displacement $\|\bar{q}^L - \bar{q}^0\|$. This step is essential to ensure unit consistency: it scales the unbounded geometric terms to a dimensionless relative range (typically comparable to unity), making them commensurable with the entropy-based difficulty term $\mathcal{H}$ and avoiding potentially incompatible units—hidden-state L2 norms and log-probabilities. We experimentally validate the necessity of this normalization and component calibration in Section 3.4.

## 3. Experiments

In this section, we conduct extensive experiments to address the following research questions: (**RQ1**) How does `GeoR` compare against existing baselines across diverse reasoning tasks and model architectures? (**RQ2**) Does `GeoR` introduce significant computational overhead? (**RQ3**) How sensitive is `GeoR` to its key components and hyperparameters?

### 3.1. Experimental Setup

**Assessment Criteria.** The central question is whether `GeoR` provides a reliable basis for self-evaluation. Intuitively, the `GeoR` score should reflect the correctness of model outputs: trajectories with high `GeoR` should correspond to correct answers with high probability, while low `GeoR` should indicate a high likelihood of incorrectness. Conversely, if `GeoR` fails to align with correctness—e.g., assigning high scores to wrong answers—then the proposed modeling is unsatisfactory. Evaluating this alignment can be cast as a ranking problem. As illustrated in Figure 7 (left), the fewer incorrect samples (red) that appear in the high-score region, the stronger the self-evaluation capability. Technically, this alignment can be quantified by standard evaluation metrics. In this paper, we adopt three widely used metrics—AUROC, AUPR, and FPR@95—to measure how well `GeoR` satisfies the self-evaluation criterion.

❶ **AUROC:** captures global ranking quality. A higher AUROC indicates that `GeoR` consistently assigns higher values to trajectories leading to correct answers, reflecting strong overall discriminative power.

❷ **AUPR:** emphasizes precision under class imbalance. It measures whether `GeoR` can effectively prioritize correct trajectories when correct cases are relatively sparse. A higher AUPR reflects stronger ability to accurately isolate reliable reasoning in imbalanced settings.

❸ **FPR@95:** evaluates robustness under high recall. It reports the false positive rate when 95% of correct trajectories are recalled. Lower FPR@95 indicates that `GeoR` achieves reliable self-evaluation with minimal overestimation of incorrect trajectories, even under stringent correctness requirements.

These metrics are widely adopted for ranking-based binary classification, particularly under imbalanced or high-recall conditions. Together, they provide a rigorous and multi-faceted evaluation of how faithfully the `GeoR` score reflects the correctness of model outputs. Formal definitions are provided in Appendix A.7.

**Baselines.** We compare against eleven baselines covering both white-box and black-box methods. For a detailed discussion, see Section A.2 and Section A.3. These baselines include `Random`, `Maximum Softmax Probability` (Wang et al., 2025a), `Perplexity` (Si et al., 2023), `Entropy` (Huang et al., 2025b), `Verbal Self-Evaluation` (Li et al., 2024), `CoE-R/C` (Wang et al., 2025a), `Internal Consistency` (Xie et al., 2024), `Forking Tokens` (Bigelow et al., 2025), `Length-normalized Entropy` (Malinin & Gales, 2021), and `Temperature Scaling` (Shih et al., 2023). For all methods, we compute the three evaluation metrics introduced above. In particular, the `Random` baseline serves as a lower bound, corresponding to a uniform ranking with no self-evaluation capability.

**Models and Benchmarks.** We validate the universality of `GeoR` across diverse model families, modalities, and reasoning training paradigms (SFT, RL, and Distillation). Specifically, we evaluate the proposed `GeoR` on seven state-of-the-art LRMs spanning a wide range of scales: `DeepSeek-R1-Distill-Qwen-1.5B/7B/14B/32B`, `DeepSeek-R1-Distill-Llama-8B/70B` (DeepSeek-AI, 2025), and `QwQ-32B` (Team, 2025c). In addition, we conduct experiments on `Qwen2.5-VL` (Bai et al., 2025) trained with GRPO to evaluate multimodal reasoning scenarios. Together, these models span diverse scales and modalities, offering a comprehensive testbed for examining the generality and robustness of `GeoR`. Detailed model architectures and training configurations are provided in Appendix A.6.

To assess generality, we experiment on seven widely adopted benchmarks. MMLU and MMLU-Pro cover multi-domain reasoning across a broad range of subjects. GSM8K, TheoremQA, and CommonsenseQA probe distinct reasoning skills, including arithmetic problem solving, formal theorem reasoning, and commonsense understanding. We further evaluate cross-lingual robustness on MGSM, considering English, German, French, and Chinese subsets. Finally, we test multimodal reasoning with Geometry3K (Lu et al., 2021), which requires spatial understanding from both textual and diagrammatic inputs. Comprehensive dataset details, including task formulations, prompts, and examples, are provided in Appendix A.5.

### 3.2. Main Results (RQ1)

Tables 1 to 10 and Figures 4 to 6 comprehensively report the performance of `GeoR` compared to eleven baselines across diverse benchmarks and model backbones. We summarize the key observations as follows:

**Takeaway ❶: `GeoR` demonstrates superior performance**

*Table 1.* Main results on GSM8K, CommonsenseQA, and TheoremQA. We categorize baselines into Black-Box (top) and White-Box (middle) methods. Arrows indicate direction of improvement.

| Model | Method | GSM8K | | | CommonsenseQA | | | TheoremQA | | |
|---|---|---|---|---|---|---|---|---|---|---|
| | | AUROC ↑ | AUPR ↑ | FPR@95 ↓ | AUROC ↑ | AUPR ↑ | FPR@95 ↓ | AUROC ↑ | AUPR ↑ | FPR@95 ↓ |
| R1-Qwen-1.5B | Random | 50.23 | 75.39 | 94.29 | 50.03 | 46.81 | 93.14 | 49.75 | 40.35 | 92.81 |
| | Verbal | 51.48 | 75.11 | 96.34 | 52.01 | 45.87 | 91.81 | 51.51 | 39.94 | 93.22 |
| | Forking | 65.19 | 83.46 | 83.65 | 59.47 | 56.38 | 86.15 | 53.18 | 42.48 | 92.47 |
| | MaxProb | 70.34 | 87.97 | 76.95 | 64.06 | 61.16 | 81.42 | 53.65 | 43.91 | 92.19 |
| | PPL | 70.88 | 88.24 | 77.29 | 63.86 | 61.07 | 83.19 | 54.05 | 44.24 | 91.25 |
| | Entropy | 71.86 | 88.54 | 75.93 | 64.26 | 61.32 | 82.30 | 54.33 | 44.52 | 92.19 |
| | CoE-R | 74.17 | 89.99 | 75.93 | 60.03 | 56.37 | 85.18 | 53.33 | 44.18 | 93.44 |
| | CoE-C | 73.32 | 89.37 | 75.25 | 58.90 | 55.44 | 85.18 | 52.42 | 43.34 | 90.31 |
| | Norm. Ent. | 70.65 | 88.18 | 77.58 | 63.72 | 60.94 | 83.47 | 54.12 | 44.31 | 91.68 |
| | Temp. Scal. | 71.14 | 88.31 | 76.84 | 64.08 | 61.19 | 82.95 | 54.18 | 44.36 | 91.52 |
| | Int. Cons. | 58.47 | 79.32 | 90.18 | 55.28 | 50.14 | 88.92 | 52.34 | 41.27 | 92.84 |
| | **GeoR** | **77.28** | **91.42** | **72.20** | **64.26** | **61.34** | **81.19** | **54.37** | **44.82** | **90.00** |
| R1-LLaMA-8B | Random | 50.22 | 81.21 | 95.33 | 51.75 | 77.22 | 96.14 | 46.29 | 41.67 | 92.91 |
| | Verbal | 51.22 | 85.49 | 94.77 | 51.04 | 75.95 | 97.41 | 49.11 | 40.24 | 93.57 |
| | Forking | 57.39 | 85.71 | 94.61 | 66.83 | 85.74 | 78.15 | 56.72 | 49.18 | 87.94 |
| | MaxProb | 58.85 | 85.46 | 94.49 | 74.71 | 89.29 | 67.38 | 59.39 | 53.12 | 84.65 |
| | PPL | 58.94 | 85.51 | 94.49 | 74.55 | 89.23 | 67.81 | 60.01 | 53.91 | 85.04 |
| | Entropy | 60.25 | 85.89 | 94.88 | 74.68 | 89.28 | 65.24 | 60.27 | 54.06 | 84.65 |
| | CoE-R | 63.01 | 86.57 | 86.22 | 52.00 | 77.92 | 93.56 | 50.56 | 46.02 | 96.46 |
| | CoE-C | 63.86 | 86.50 | 86.61 | 73.90 | 88.92 | 69.96 | 58.01 | 54.10 | 93.70 |
| | Norm. Ent. | 59.17 | 85.63 | 94.72 | 74.48 | 89.18 | 68.13 | 60.08 | 53.97 | 84.89 |
| | Temp. Scal. | 59.42 | 85.71 | 94.35 | 74.62 | 89.26 | 67.56 | 60.15 | 54.02 | 84.78 |
| | Int. Cons. | 54.18 | 85.63 | 94.25 | 58.42 | 81.36 | 89.73 | 52.64 | 44.91 | 91.28 |
| | **GeoR** | **68.12** | **89.37** | **80.31** | **74.71** | **89.35** | **65.24** | **60.57** | **56.06** | **83.86** |
| R1-Qwen-14B | Random | 49.46 | 93.23 | 95.24 | 49.13 | 84.30 | 95.03 | 50.93 | 56.79 | 94.93 |
| | Verbal | 50.44 | 91.91 | 96.14 | 54.19 | 85.87 | 94.85 | 51.83 | 56.54 | 94.72 |
| | Forking | 53.76 | 92.84 | 89.27 | 71.56 | 91.83 | 78.62 | 52.41 | 57.62 | 91.75 |
| | MaxProb | 53.98 | 93.06 | 85.71 | 78.90 | 94.87 | 65.22 | 51.94 | 57.71 | 89.40 |
| | PPL | 54.51 | 93.23 | 84.52 | 79.24 | 94.99 | 64.60 | 52.39 | 58.07 | 89.40 |
| | Entropy | 55.13 | 93.26 | 85.71 | 79.24 | 94.97 | 65.84 | 52.62 | 58.24 | 90.32 |
| | CoE-R | 59.62 | 93.84 | 77.38 | 71.78 | 91.82 | 73.29 | 52.28 | 58.32 | 93.55 |
| | CoE-C | 58.63 | 93.66 | 77.38 | 68.40 | 90.78 | 80.75 | 50.99 | 57.39 | 92.63 |
| | Norm. Ent. | 54.63 | 93.21 | 84.78 | 79.18 | 94.96 | 64.89 | 52.45 | 58.11 | 89.68 |
| | Temp. Scal. | 54.89 | 93.25 | 85.14 | 79.26 | 94.98 | 65.37 | 52.53 | 58.17 | 89.95 |
| | Int. Cons. | 51.93 | 92.38 | 93.86 | 62.47 | 89.15 | 86.34 | 52.08 | 56.97 | 93.18 |
| | **GeoR** | **61.64** | **94.04** | **76.19** | **79.27** | **95.00** | **61.49** | **52.65** | **58.77** | **88.02** |

**across diverse architectures and domains.** As shown in Table 1 and Table 8, GeoR consistently outperforms baselines across diverse tasks (arithmetic, commonsense, symbolic) and model families. This counters the concern that geometric metrics might merely capture architectural transformations or incidental activation scaling rather than true reasoning dynamics. Instead, the robust correlation between GeoR and reasoning correctness—persisting across varied difficulty levels and datasets—suggests it captures intrinsic indicators of reasoning fidelity, dispelling doubts about its generalizability beyond specific experiments.

**Takeaway ❷:** GeoR **exhibits strong scalability across varied model sizes (Figure 5).** In Table 9, we further validate

performance on heterogeneous LRMs ranging from 1.5B to 70B parameters. GeoR consistently achieves leading performance, indicating that the geometric signal it captures is not an artifact of small-scale models but a scalable property of the reasoning process in large language models.

**Takeaway ❸:** GeoR **generalizes effectively to multilingual and multimodal contexts (Figure 4).** Experiments on MGSM (Table 10) confirm robust performance across diverse languages (English, German, French, Chinese). Furthermore, we validate GeoR in multimodal scenarios using Qwen2.5-VL on Geometry3K within the Easy-R1 framework (Team, 2025a). As shown in Table 3, GeoR consistently outperforms baselines even when reasoning involves

## Multimodal     Multilingual Reasoning     Open World     Knowledge Reasoning

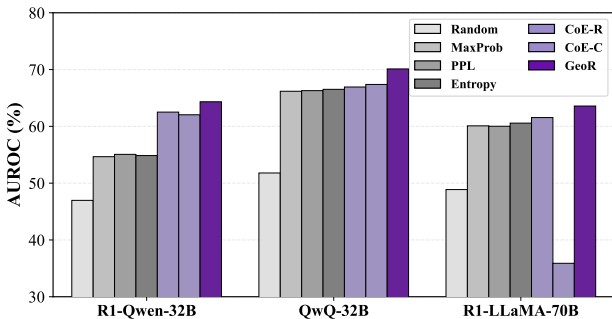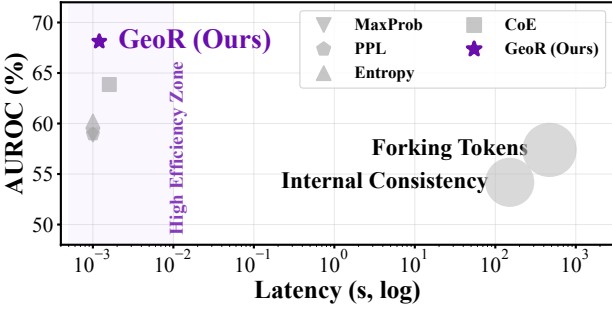

*Figure 4.* Holistic Evaluation of `GeoR` across Four Key Dimensions. The radar charts visualize the performance superiority of `GeoR` against baselines across: (1) **Multimodal capabilities** on Geometry3K (AUROC, AUPR, FPR@95); (2) **Multilingual robustness** on MGSM (English, German, French, Chinese); (3) **Open-World generalization** under distribution shifts; and (4) **Knowledge Reasoning** on MMLU across varied model scales (1.5B to 14B). `GeoR` consistently encompasses the largest area, demonstrating superior generalizability and stability compared to baselines.

*Figure 5.* **Performance Scalability of `GeoR`.** We compare AU-ROC scores on GSM8K across `R1-Qwen-32B`, `QwQ-32B`, and `R1-LLaMA-70B`. `GeoR` (purple) consistently outperforms base-line methods across all model architectures, confirming that the geometric signal remains discriminative even as model scale increases.

*Figure 6.* Performance (AUROC) vs. Execution Latency trade-off on GSM8K (R1-LLaMA-8B). `GeoR` (top-left) achieves the best performance while maintaining millisecond-level efficiency, an order of magnitude faster than sampling-based baselines.

integrating complex textual and visual signals. These results indicate that the geometric evolution of hidden states serves as a universal proxy for reasoning quality, independent of the specific input modality or linguistic surface form.

**Takeaway ❹: `GeoR` is stable under Open-World Distributional Shifts (Figure 4).** To stress-test generalizability, we constructed a mixed-distribution "open-world" setting: (1) *Mixed Math* (GSM8K + TheoremQA) and (2) *Mixed Reasoning* (CommonsenseQA + MMLU), balancing samples to avoid size bias. As illustrated in Table 5, `GeoR` clearly outperforms all baselines on both mixed distributions, while random guessing remains near 50%, demonstrating that its behavior is robust to heterogeneous input distributions—an environment where incidental geometric artifacts would likely collapse.

### 3.3. Cost Analysis (RQ2)
To evaluate the efficiency of `GeoR`, we analyze its computa-tional cost profile (visualized in Figure 6).

**Takeaway ❺: `GeoR` achieves superior efficiency with neg-**

ligible latency. As demonstrated in Table 6, `GeoR` incurs only millisecond-level overhead ($1.2 \times 10^{-3}$ s), which is negligible relative to a single inference pass (15.24 s). In sharp contrast, baselines such as Internal Consistency and Forking Tokens suffer from prohibitively high computa-tional costs (151.14 s and 471.36 s, respectively) due to the requirement for multiple sampling and reasoning paths. This confirms that `GeoR` is not only effective but also ex-ceptionally resource-friendly, making it highly suitable for real-world self-evaluation deployment.

### 3.4. Framework Analysis (RQ3)

**Ablation Study.** We ablate the key design choices in `GeoR` from the following perspectives:

① **Interpretation of Evolution Dimension (Layer-wise vs. Token-wise):** A critical distinction in our framework is that our *Geometric Evolution* operates along the *layer dimension* (depth), rather than the temporal token sequence. We ex-plicitly compare our layer-wise modeling with a token-wise variant (tracking changes between successive CoT tokens). Our results reveal that token-wise dynamics perform poorly (often close to random baseline), as they are dominated

*Table 2.* Ablation study of trajectory modeling strategies across 4 models on GSM8K. `GeoR` (Mean Reasoning) consistently outperforms token-wise and alternative pooling methods.

| Strategy | R1-Qwen-1.5B | | | R1-Qwen-7B | | | R1-LLaMA-8B | | | R1-Qwen-14B | | |
|---|---|---|---|---|---|---|---|---|---|---|---|---|
| | AUROC ↑ | AUPR ↑ | FPR@95 ↓ | AUROC ↑ | AUPR ↑ | FPR@95 ↓ | AUROC ↑ | AUPR ↑ | FPR@95 ↓ | AUROC ↑ | AUPR ↑ | FPR@95 ↓ |
| Mean Reasoning ✔ | **77.28** | **91.42** | **72.20** | **76.32** | **94.73** | **67.27** | **68.12** | **89.37** | **80.31** | **61.64** | **94.04** | **76.19** |
| Last CoT token | 71.54 | 90.12 | 76.77 | 71.94 | 93.61 | 73.33 | 63.44 | 88.14 | 86.54 | 59.27 | 92.58 | 81.28 |
| Mean of all output tokens | 70.53 | 88.56 | 79.83 | 70.90 | 92.96 | 75.76 | 61.43 | 86.57 | 85.42 | 55.64 | 91.79 | 79.58 |
| Layer-wise Evolution ✔ | **77.28** | **91.42** | **72.20** | **76.32** | **94.73** | **67.27** | **68.12** | **89.37** | **80.31** | **61.64** | **94.04** | **76.19** |
| Token-wise Dynamics | 51.84 | 73.92 | 96.17 | 52.45 | 78.44 | 90.12 | 48.93 | 82.64 | 96.88 | 51.17 | 91.86 | 96.52 |

*Table 3.* Ablation study of `GeoR` Reasoning Score Components

| $D_1$ | $D_2$ | $\mathcal{H}$ | AUROC ↑ | AUPR ↑ | FPR@95 ↓ |
|---|---|---|---|---|---|
| ✓ | | | 74.35 | 92.60 | 72.12 |
| | ✓ | | 74.07 | 92.58 | 73.33 |
| ✓ | ✓ | | 75.23 | 93.60 | 72.73 |
| ✓ | | ✓ | 69.71 | 93.00 | 78.18 |
| | ✓ | ✓ | 70.11 | 93.07 | 78.18 |
| | w/o Norm | | 75.10 | 93.52 | 73.05 |
| ✓ | ✓ | ✓ | **76.32** | **94.73** | **67.27** |

*Table 4.* Robustness analysis on GSM8K (DeepSeek-R1-Distill-Qwen-7B) over 3 runs (Mean ± Std). `GeoR` shows the best stability.

| Method | AUROC | AUPR | FPR@95 |
|---|---|---|---|
| MaxProb | $67.5 \pm 2.0$ | $92.0 \pm 1.9$ | $82.1 \pm 2.2$ |
| PPL | $67.0 \pm 1.7$ | $92.6 \pm 2.2$ | $82.8 \pm 1.8$ |
| Entropy | $68.5 \pm 2.1$ | $92.3 \pm 1.8$ | $79.1 \pm 2.0$ |
| CoE-R | $71.3 \pm 1.9$ | $93.1 \pm 2.0$ | $74.2 \pm 1.9$ |
| CoE-C | $70.6 \pm 2.2$ | $92.9 \pm 1.7$ | $74.9 \pm 2.1$ |
| `GeoR` | $\mathbf{76.8 \pm 1.6}$ | $\mathbf{94.5 \pm 1.5}$ | $\mathbf{67.0 \pm 1.4}$ |

by local lexical shifts (e.g., "red" → "apple") rather than meaningful reasoning progress. In contrast, our layer-wise geometric evolution captures the iterative refinement of the reasoning state along the network depth, providing a far more robust and interpretable proxy for the intrinsic quality of the model's thinking process.

② **Reasoning Token Aggregation Strategies:** We examine three strategies to construct the trajectory representation: (i) *Mean Reasoning* (averaging hidden states over reasoning tokens), (ii) *Last Token* (using only the final CoT token), and (iii) *Global Mean* (averaging over the entire output including conclusion). Theoretically, mean pooling acts as a semantic denoiser, preserving the global semantic signal of the reasoning trajectory that is most relevant for self-evaluation, while suppressing token-level noise and idiosyncrasies. Empirically, as shown in Table 2, *Mean Reasoning* consistently outperforms other methods, improving AUROC by 3–7 points and significantly lowering FPR@95. In contrast, *Last Token* approaches lose the evolutionary history, and *Global Mean* dilutes the geometric signal with non-reasoning tokens. This confirms that the average dynamics of the *reasoning process itself* provide the most faithful proxy for determining solution correctness.

③ **Impact of `GeoR` Components:** We disentangle the contribution of Geometric Evolution ($D_1$, $D_2$) and Difficulty-Aware Calibration ($\mathcal{H}$) in Table 3. Using Geometric Evolution ($D_1$ or $D_2$) alone yields suboptimal performance, as it overlooks the intrinsic difficulty of the query—a rapidly evolving trajectory might indicate confusion rather than progress in simpler tasks. Introducing Difficulty-Aware Calibration ($\mathcal{H}$) normalizes this evolution, effectively penalizing uncertain paths. Additionally, the "w/o Norm" entry in Table 3 reveals that omitting the trajectory displacement

normalization leads to a noticeable performance drop (AUROC 76.32% → 75.10%). This substantiates the necessity of the normalization design in Equation (4): without normalization, the geometric terms exhibit significant scale discrepancies incompatible with the calibration term, thereby affecting the final measurement. Furthermore, to avoid introducing additional complexity, we refrain from setting extra hyperparameters for fine-tuning, as the current parameter-free normalization already yields valid performance. The combination ($D_1 + D_2 - \mathcal{H}$) thus achieves the highest performance by successfully distinguishing decisive, efficient reasoning from chaotic or uncertain generation.

**Sensitivity & Statistical Significance.** To bolster the reliability and transparency of our experimental claims, we rigorously evaluate the statistical significance of our results by reporting standard deviations across multiple runs. Specifically, we reproduced experiments on GSM8K using three semantically equivalent prompts with DeepSeek-R1-Distill-Qwen-7B. As presented in Table 4, `GeoR` not only achieves superior mean performance but also exhibits significantly lower variance ($\pm 1.6$) compared to baselines (e.g., up to $\pm 2.2$). This demonstrates that our improvements are statistically robust and that `GeoR` captures intrinsic reasoning signals distinctive from random fluctuations.

## 4. Conclusion

We introduced `GeoR`, a white-box framework quantifying reasoning fidelity via layerwise trajectory evolution and difficulty calibration. Extensive experiments across varied models and tasks confirm that `GeoR` consistently outperforms baselines, establishing it as a robust, interpretable tool for trustworthy LRM self-evaluation.

## Impact Statement

This work introduces GeoR (Geometry of Reasoning), a white-box self-evaluation framework designed to enhance the reliability and transparency of Large Reasoning Models (LRMs). The potential societal and academic impacts of this research include:

- **Enhancing AI Reliability and Safety:** By leveraging layerwise trajectory evolution to distinguish coherent reasoning from chaotic errors, GeoR enables models to internally verify their outputs without external supervision. This is critical for reducing hallucinations and ensuring the safety of AI-driven decision-making in sensitive domains.

- **Advancing Model Interpretability:** Unlike black-box methods that offer limited insight into a model's "confidence," GeoR provides a principled geometric perspective on the reasoning process. By quantifying the first- and second-order evolution of hidden states, it offers a window into the "depth of thought" and internal refinement of representations.

- **Resource-Friendly Deployment:** GeoR achieves superior performance with negligible computational overhead (millisecond-level latency), making it significantly more efficient than mainstream sampling-based self-evaluation methods. This efficiency supports the sustainable and real-time monitoring of large-scale AI systems.

- **Broad Generalizability:** Our findings demonstrate that geometric signals of reasoning fidelity are robust across diverse model architectures, languages, and modalities (including vision-language reasoning). This ensures that reliability-enhancing techniques can be applied equitably across different linguistic and task contexts.

## Acknowledgements

The authors gratefully acknowledge the scientific support and HPC resources provided by the Erlangen National High Performance Computing Center (NHR@FAU) of the Friedrich-Alexander-Universität Erlangen-Nürnberg (FAU) under the NHR project b270dd and b271dd. NHR funding is provided by federal and Bavarian state authorities. NHR@FAU hardware is partially funded by the German Research Foundation (DFG) – 440719683.

We acknowledge the EuroHPC Joint Undertaking for awarding this project access to the EuroHPC supercomputer LEONARDO, hosted by CINECA (Italy) and the LEONARDO consortium through an EuroHPC AI and Data-Intensive Applications Access call EHPC-AI-2024A06-060.

This work is partially supported by the NTU AI-for-X Post-doctoral Fellowship.

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

# A. Appendix

## A.1. Detailed Experimental Results

In this section, we provide a comprehensive analysis of `GeoR` across diverse benchmarks, model scales, and modalities. We detail the experimental setup and discuss the results presented in Tables 6, 5, 7, 8, 9, and 10.

**Robustness under Open-World Distributional Shift.** Real-world deployments often face heterogeneous data distributions. To simulate this, we constructed two open-world settings: *Mixed Math* (combining GSM8K and TheoremQA) and *Mixed Reasoning* (combining CommonsenseQA and MMLU). As detailed in Table 5, `GeoR` demonstrates exceptional stability, significantly creating a performance gap over baselines. For instance, on the *Mixed Reasoning* task with R1-LLaMA-8B, `GeoR` achieves an AUROC of 70.42%, surassing the strongest white-box baseline (CoE-C at 67.02%) by over 3%. This confirms that `GeoR`'s geometric signal is robust to varying input distributions and difficulty levels.

*Table 5.* Performance (AUROC) under Open-World Distributional Shift.

| Model | Method | Mixed Math | Mixed Reasoning |
|---|---|---|---|
| R1-Qwen -1.5B | Random | 47.12 | 50.18 |
| | Verbal | 53.48 | 51.11 |
| | MaxProb | 59.73 | 61.92 |
| | Entropy | 60.25 | 63.04 |
| | CoE-R | 61.02 | 64.23 |
| | CoE-C | 61.35 | 64.57 |
| | **GeoR** | **65.21** | **68.03** |
| R1-LLaMA -8B | Random | 51.09 | 48.07 |
| | Verbal | 49.02 | 52.89 |
| | MaxProb | 60.67 | 64.35 |
| | Entropy | 61.84 | 65.41 |
| | CoE-R | 62.41 | 66.71 |
| | CoE-C | 63.07 | 67.02 |
| | **GeoR** | **66.82** | **70.42** |

**Computational Efficiency Analysis.** Efficiency is paramount for self-evaluation methods. Table 6 compares the execution overhead of `GeoR` against sampling-based approaches on GSM8K using R1-LLaMA-8B. While methods like *Internal Consistency* and *Forking Tokens* require prohibitively expensive multiple forward passes (151.14s and 471.36s, respectively), `GeoR` incurs a negligible overhead of just $1.2 \times 10^{-3}$ seconds. This near-zero latency, combined with superior performance, positions `GeoR` as an ideal candidate for real-time monitoring and rejection sampling.

*Table 6.* Execution time comparison on GSM8K with R1-LLaMA-8B (NVIDIA H100). `GeoR` achieves rapid evaluation without noticeable latency.

| Method | Execution Time (s) |
|---|---|
| Inference (1 sample) | 15.24 |
| + Internal Consistency | 151.14 |
| + Forking Tokens | 471.36 |
| CoE | $1.6 \times 10^{-3}$ |
| **GeoR** | $\mathbf{1.2 \times 10^{-3}}$ |

**Multimodal Reasoning on Geometry3K.** We extend our evaluation to the multimodal domain using the Geometry3K benchmark with `Qwen2.5-VL`. Table 7 reports the results. `GeoR` achieves an AUROC of 69.41%, outperforming Entropy (66.21%) and MaxProb (64.21%). Notably, `GeoR` also secures the lowest FPR@95 (82.50%), indicating its reliability in identifying incorrect multimodal reasoning chains involving both text and geometric diagrams.

**Complex Reasoning on MMLU and MMLU-Pro.** Table 8 presents the performance on general knowledge and complex reasoning tasks. On MMLU-Pro, a more challenging version of MMLU, `GeoR` consistently ranks first across all model sizes. For R1-Qwen-14B, it achieves an AUROC of 65.89% and AUPR of 48.57%, significantly outperforming the Entropy baseline (66.12% AUROC, but lower AUPR of 48.31% and higher FPR). The gains are equally pronounced on standard MMLU, where `GeoR` reaches 79.13% AUROC with R1-Qwen-14B, showcasing its versatility beyond pure mathematical reasoning.

**Scalability to Large Reasoning Models.** We investigate whether `GeoR` scales to larger architectures in Table 9. Testing on `R1-Qwen-32B`, `QwQ-32B`, and `R1-LLaMA-70B`, we observe that `GeoR` maintains its leadership. On `QwQ-32B`, it achieves a remarkable 70.11% AUROC and 97.79% AUPR. Crucially, the performance gap remains significant even at 70B parameters, suggesting that the geometric evolution of reasoning trajectories is a fundamental property of LRMs that does not diminish with scale.

**Multilingual Generalization on MGSM.** Finally, we assess cross-lingual robustness on the MGSM benchmark (Table 10). `GeoR` consistently outperforms baselines across English, German, French, and Chinese. For instance, in Chinese (zh), `GeoR` achieves 71.04% AUROC compared to 65.14% for Entropy. The consistency across languages implies that `GeoR` captures the language-agnostic "quality of thought" rather than language-specific surface features.

*Table 7.* Self-evaluation on Geometry3K.

| Method | AUROC ↑ | AUPR ↑ | FPR@95 ↓ |
|--------|---------|--------|----------|
| Random | 51.35 | 44.63 | 95.78 |
| MaxProb | 64.21 | 53.82 | 85.61 |
| PPL | 61.78 | 51.95 | 86.11 |
| Entropy | 66.21 | 55.61 | 87.24 |
| CoE-R | 69.13 | 59.60 | 82.80 |
| CoE-C | 57.42 | 46.73 | 90.50 |
| `GeoR` | **69.41** | **60.00** | **82.50** |

*Table 8.* Performance comparison on MMLU and MMLU-Pro, two multi-task benchmarks requiring both world knowledge and complex reasoning.

| Dataset | Method | R1-Qwen-1.5B | | | R1-Qwen-7B | | | R1-LLaMA-8B | | | R1-Qwen-14B | | |
|---------|--------|---------|--------|----------|---------|--------|----------|---------|--------|----------|---------|--------|----------|
| | | AUROC ↑ | AUPR ↑ | FPR@95 ↓ | AUROC ↑ | AUPR ↑ | FPR@95 ↓ | AUROC ↑ | AUPR ↑ | FPR@95 ↓ | AUROC ↑ | AUPR ↑ | FPR@95 ↓ |
| | Random | 49.51 | 25.83 | 95.72 | 50.71 | 30.83 | 94.96 | 50.95 | 31.25 | 94.30 | 50.22 | 40.12 | 94.31 |
| | MaxProb | 61.44 | 33.40 | 91.05 | 60.11 | 35.55 | 91.87 | 59.29 | 36.68 | 90.54 | 65.54 | 48.33 | 85.52 |
| | PPL | 61.65 | 33.38 | 90.89 | 60.31 | 35.86 | 90.96 | 59.42 | 36.86 | 90.44 | 65.34 | 47.98 | 86.23 |
| MMLU-Pro | Entropy | 62.28 | 33.93 | 90.57 | 59.89 | 34.89 | 90.10 | 59.82 | 37.12 | 89.93 | 66.12 | 48.31 | 85.45 |
| | CoE-R | 57.78 | 30.76 | 91.25 | 52.63 | 31.34 | 92.81 | 51.77 | 33.66 | 94.12 | 61.45 | 45.12 | 86.77 |
| | CoE-C | 54.50 | 28.45 | 92.95 | 55.41 | 32.89 | 92.13 | 57.89 | 36.20 | 89.60 | 61.52 | 46.53 | 87.10 |
| | `GeoR` | **62.46** | 33.39 | **90.41** | **61.89** | **36.13** | **89.45** | **60.47** | **37.90** | **89.24** | 65.89 | **48.57** | **85.12** |
| | Random | 50.22 | 57.23 | 95.80 | 48.27 | 70.83 | 95.42 | 50.29 | 78.96 | 95.75 | 50.65 | 85.72 | 94.69 |
| | MaxProb | 64.61 | 75.37 | 91.81 | 72.11 | 87.04 | 84.77 | 71.29 | 89.55 | 82.54 | 78.20 | 93.77 | 71.69 |
| | PPL | 65.19 | 75.50 | 90.96 | 72.33 | 87.22 | 84.36 | 71.32 | 89.68 | 83.25 | 78.26 | 94.03 | 72.28 |
| MMLU | Entropy | 66.40 | 76.38 | 89.95 | 73.23 | 87.65 | 83.62 | 71.87 | 89.87 | 83.16 | 78.17 | 93.73 | 72.42 |
| | CoE-R | 58.79 | 65.18 | 93.20 | 64.94 | 80.61 | 85.79 | 49.49 | 80.25 | 95.31 | 71.59 | 92.94 | 82.95 |
| | CoE-C | 56.34 | 62.21 | 93.97 | 62.74 | 79.32 | 88.27 | 66.34 | 86.60 | 85.83 | 69.34 | 92.39 | 87.02 |
| | `GeoR` | **68.14** | **76.47** | 90.36 | **74.26** | **88.18** | 84.39 | **73.49** | **90.71** | 82.59 | **79.13** | **94.84** | 72.12 |

## A.2. Related Work

**Large Reasoning Models (LRMs):** Comparing with LLMs, LRMs have demonstrated that extending the reasoning process can substantially enhance model inference performance (Cui et al., 2025; Yuan et al., 2025a). For example, `OpenAI-o1` (OpenAI, 2024), `Qwen-QwQ` (Team, 2025c) and `DeepSeek-R1` (DeepSeek-AI, 2025) explicitly perform CoT reasoning by constructing an intermediate reasoning trajectory trying to justify the final answer (Wei et al., 2022). LRMs show to be effective in domains requiring complex logical deduction, such as mathematics and programming (Zhao

*Table 9.* GSM8K results on larger models (Higher AUROC and AUPR indicate better performance; Lower FPR@95 indicates better performance).

| Dataset | Method | R1-Qwen-32B | | | QwQ-32B | | | R1-LLaMA-70B | | |
|---|---|---|---|---|---|---|---|---|---|---|
| | | AUROC ↑ | AUPR ↑ | FPR@95 ↓ | AUROC ↑ | AUPR ↑ | FPR@95 ↓ | AUROC ↑ | AUPR ↑ | FPR@95 ↓ |
| GSM8K | Random | 46.97 | 94.05 | 93.59 | 51.79 | 96.55 | 93.88 | 48.87 | 94.32 | 97.22 |
| | MaxProb | 54.66 | 94.88 | 94.81 | 66.18 | 97.07 | 72.34 | 60.09 | 96.14 | 85.71 |
| | PPL | 55.07 | 94.94 | 93.51 | 66.29 | 97.12 | 72.34 | 60.02 | 96.13 | 85.71 |
| | Entropy | 54.86 | 94.88 | 93.51 | 66.52 | 97.32 | 70.21 | 60.57 | 96.24 | 85.71 |
| | CoE-R | 62.51 | 96.62 | 96.10 | 66.93 | 97.64 | 85.11 | 61.55 | 96.33 | 88.57 |
| | CoE-C | 62.04 | 96.49 | 96.10 | 67.38 | 97.69 | 87.23 | 35.89 | 92.45 | 98.57 |
| | GeoR | **64.33** | **96.75** | 96.10 | **70.11** | **97.79** | **70.21** | **63.58** | 96.23 | **85.71** |

*Table 10.* Results on MGSM dataset across four languages using DeepSeek-R1-Distill-Qwen-7B (Arrows indicate direction of improvement).

| Model | Method | English (en) | | | German (de) | | | French (fr) | | | Chinese (zh) | | |
|---|---|---|---|---|---|---|---|---|---|---|---|---|---|
| | | AUROC ↑ | AUPR ↑ | FPR@95 ↓ | AUROC ↑ | AUPR ↑ | FPR@95 ↓ | AUROC ↑ | AUPR ↑ | FPR@95 ↓ | AUROC ↑ | AUPR ↑ | FPR@95 ↓ |
| DeepSeek-R1-Distill-Qwen-7B | Random | 54.12 | 91.52 | 100.00 | 45.08 | 74.66 | 96.83 | 52.67 | 78.79 | 95.12 | 49.42 | 68.94 | 95.89 |
| | MaxProb | 70.95 | 95.00 | 72.00 | 69.15 | 86.51 | 80.95 | 70.63 | 82.19 | 73.97 | 62.76 | 89.03 | 95.12 |
| | PPL | 70.92 | 94.98 | 72.00 | 69.36 | 86.62 | 80.95 | 70.91 | 82.86 | 73.97 | 63.25 | 89.01 | 92.68 |
| | Entropy | 73.24 | 95.40 | 68.00 | 69.15 | 87.04 | 80.95 | 70.27 | 82.04 | 72.60 | 65.14 | 90.13 | 92.68 |
| | CoE-R | 67.75 | 94.36 | 72.00 | 60.56 | 79.02 | 80.95 | 55.17 | 70.96 | 94.52 | 64.45 | 89.11 | 92.68 |
| | CoE-C | 66.58 | 94.04 | 72.00 | 58.18 | 78.08 | 85.71 | 54.15 | 70.21 | 98.63 | 63.43 | 88.61 | 95.12 |
| | GeoR | **74.03** | **95.27** | **64.00** | **69.44** | 86.91 | **79.37** | **71.32** | 82.36 | **72.60** | **71.04** | **91.28** | **92.68** |

et al., 2025; Bi et al., 2025; Chen et al., 2025b; Wang et al., 2026; Rong et al., 2026; Chen et al., 2025a; Peng et al., 2025; Zhang et al., 2023; Ma et al., 2025; Tian et al., 2025; Lu et al., 2025; Huang et al., 2025a; Wan et al., 2025b;a; Li et al., 2026; Yang et al., 2026). Unlike conventional LLMs, which produce an answer in a single forward pass guided by shallow heuristics, LRMs follow an explicit "think-then-answer" routine (DeepSeek-AI, 2025; OpenAI, 2024; Team, 2025c;b; Muennighoff et al., 2025). They first enclose a CoT segment between special tokens (e.g., <think>...</think>) and repeatedly update internal representations while attending to these intermediate tokens; They generate the final answer only after completing the step-by-step deliberation process.

Current efforts to evaluate the overall output quality of an LRM can be broadly categorized into external and internal assessment methods, which are separately reviewed as follows.

**External Assessment:** This type of assessment methods relies on supervision signals external to the model, such as linear probing techniques (Wang & Xu, 2025), additional annotations or knowledge bases (Li et al., 2023; Nguyen et al., 2024), and proxy models (He et al., 2025). For example, (Kadavath et al., 2022) leverages the instruction-following capabilities of LLMs, enhanced by RLHF (Ouyang et al., 2022), to generate confidence scores through carefully crafted prompts. This paradigm encompasses a variety of general methodologies that directly elicit confidence estimates from LLMs via multi-stage pipelines, as broadly evaluated in (Lin et al., 2022; Tian et al., 2023; Wang et al., 2024c; 2025b; 2024b). (Manakul et al., 2023) further investigates this approach by prompting GPT-3 to generate biographical passages from the WikiBio dataset and manually annotating the factuality of the generated content. Similarly, the PSA pipeline (Xiong et al., 2024) estimates confidence by perturbing prompts (Gao et al., 2024) to produce multiple outputs and subsequently measuring their consistency. However, these methods typically require access to ground-truth labels, external resources, or auxiliary models, limiting their generalizability in real-world scenarios where such information may be unavailable or impractical to obtain. This limitation motivates the exploration of a label-free, model-internal evaluation strategies.

**Internal Assessment:** This type of assessment methods leverages the internal states obtained from the model itself, such as hidden representations (Wang et al., 2025a) or uncertainty-based signals (Shih et al., 2023; Huang et al., 2025b; Si et al., 2023), to infer the reliability of its outputs without relying on external supervision. For example, (Plaut et al., 2025) uses softmax probabilities to predict the correctness of multiple-choice questions, while (Hendrycks et al., 2022) explores temperature scaling techniques for output probabilities. (Kuhn et al., 2023) and (Farquhar et al., 2024) investigate the

concept of semantic entropy, aiming to capture uncertainty beyond surface-level probabilities. Furthermore, (Duan et al., 2024) introduces a shift toward utilizing semantic information in entropy-based reliability estimation.

### A.3. Baseline Implementation Details

To provide a comprehensive evaluation, we compare against various baselines. We detail their implementations and key adaptations for self-evaluation below:

**1. Internal Consistency (IC) (Xie et al., 2024)** Originally a test-time scaling method, we adapt it for self-evaluation by measuring the consistency of reasoning paths. We sample $n = 8$ reasoning paths and perform $l$ latent-layer decodings for each path, where $l$ matches the model's layer count. The final score is a normalized combination of Self-Consistency (SC) and Internal Consistency (IC).

**2. Forking Tokens (Bigelow et al., 2025)** This method identifies "forking tokens"—positions where alternate sampling leads to divergent outcomes—to reveal uncertainty. We adapt its output-distribution drift metric, which captures uncertainty fluctuations during generation. Candidate tokens are included if their probability is $\geq 20\%$, and the drift measure is normalized to serve as a correctness score.

**3. Length-normalized Entropy (Malinin & Gales, 2021)** This baseline computes the entropy of the generated sequence, normalized by its length to account for variable-length reasoning chains. We use top-$k$ sampling ($k = 5$) to generate outputs and compute their average entropy.

**4. Temperature Scaling (Shih et al., 2023)** A classic calibration technique that adjusts the softmax distribution. We apply temperature scaling with $T = 0.7$ to calibrate the output probabilities before computing the Maximum Softmax Probability (MSP) as the confidence score.

**5. Verbal Self-Evaluation** We additionally implemented a *verbal* self-evaluation baseline, where the model is prompted to output a confidence score in $[0, 1]$ after producing its answer. This approach is attractive because it is label-free and training-free; however, in our experiments it showed limited discriminative power for reasoning quality. Concretely, scores clustered in a narrow band ($\approx 0.95$–$1.0$) for both correct and incorrect trajectories, yielding poor separation under AUROC/AUPR and high FPR@95. We hypothesize two causes: (i) verbalized confidence is systematically over-optimistic relative to ground truth, and (ii) a single prompt format induces prompt-level bias that is largely decoupled from the underlying reasoning trajectory. As a result, this baseline was not included in every comparisons; we report it here for completeness and future replication.

---

**Self-Evaluation Prompt**

**Instruction:** Solve this math problem. Give the reasoning steps before giving the final answer on the last line by itself in the format of "Answer:". Do not add anything other than the integer answer after "Answer:".

**Then:** Provide a self-evaluation score between 0 and 1 of how confident you are in the correctness of your answer. Output the score in the format: "Score: <float between 0 and 1>"

**Question:**
`{input_data}`

---

For transparency, we release the exact prompt and evaluation script; researchers can reproduce the baseline and compare verbal scores with representation-based metrics like `GeoR` under identical settings.

### A.4. Multimodal Validation on Geometry3K

We validate `GeoR` in a multimodal setting using the Geometry3K benchmark, which contains 3,002 geometry problems pairing textual statements with diagrams and formal annotations for symbolic reasoning. *Geometry3K* is introduced by Inter-GPS and targets integrated language–vision reasoning in Euclidean geometry (Lu et al., 2021). We employ **Qwen2.5-VL** (Instruct, 7B unless noted) as the vision–language backbone, following its official release notes and model repositories, which describe the series' multimodal encoder–decoder integration and available open-weight checkpoints (3B/7B/72B). Training and inference are orchestrated with **Easy-R1**, a clean fork of veRL designed for efficient, scalable, multi-modality RL/SFT workflows (with vLLM SPMD support).

For evaluation, each sample consists of the problem text, the diagram image, and multiple-choice options (A–D). We prompt the model to reason step by step and produce a single final option letter. During decoding, we log token-level hidden states over the reasoning segment and mean-pool per transformer layer to obtain $\bar{q}^i$, enabling computation of Geometric Evolution $(D_1, D_2)$ and the Difficulty-Aware Calibration term $\mathcal{H}$. We then assess how well `GeoR` separates correct from incorrect predictions using AUROC/AUPR/FPR@95 on Geometry3K, demonstrating that the score remains informative when visual signals are integrated into the reasoning trajectory.

## A.5. Benchmark Dataset Details

We evaluate the generality and effectiveness of our proposed `GeoR` across seven widely adopted reasoning benchmarks.

**GSM8K (Cobbe et al., 2021):** a dataset designed for math word problem-solving tasks, involving arithmetic calculations and logical reasoning. Each question expects an integer as the final answer, accompanied by explicit reasoning steps. We follow standard evaluation metrics and report accuracy.

**CommonsenseQA (Talmor et al., 2019):** a multiple-choice dataset evaluating commonsense reasoning. Each question provides five candidate answers (A–E), requiring the model to choose the correct option. Answers are evaluated by accuracy, emphasizing both factual knowledge and reasoning capability.

**TheoremQA (Chen et al., 2023):** a dataset used to evaluate mathematical theorem reasoning and deduction abilities. It includes questions covering diverse mathematical topics, with answers in various formats such as numerical results, Boolean values, or lists of numbers. Performance is measured via accuracy.

**MMLU (Hendrycks et al., 2021):** a dataset assessing models across diverse knowledge domains. It comprises multiple-choice questions from fields such as mathematics, sciences, and humanities, making it ideal for evaluating model versatility and general knowledge retrieval. Answers are evaluated using accuracy.

**MMLU-Pro (Wang et al., 2024a):** An extension of MMLU, MMLU-Pro introduces a more challenging set of questions to test robust multitask reasoning capabilities of LLMs. Similar to MMLU, questions span multiple disciplines and use accuracy as the evaluation metric.

**MGSM (Shi et al., 2023)** MGSM is a multilingual benchmark focusing on math word problems. It evaluates arithmetic reasoning capabilities across multiple languages, including English, German, French, and Chinese. We evaluate performance using accuracy on numeric answers, demonstrating multilingual applicability.

**Geometry3K (Lu et al., 2021)** Geometry3K is a large-scale geometry reasoning benchmark consisting of 3,002 geometry problems with dense annotation in formal language. Each problem combines a textual description and a geometric diagram, requiring models to perform integrated visual-textual reasoning over geometry constructions. In our experiments, we evaluate performance on Geometry3K by measuring accuracy of the extracted numeric or categorical answers, demonstrating our method's applicability to multimodal geometry reasoning tasks.

Detailed dataset statistics, task specifications, prompting templates, and illustrative examples for these benchmarks are provided in Table 11 and Table 12.

## A.6. Model Details

Table 13 summarizes the model configurations used in our experiments. We cover a range of model scales and architectures to validate the robustness and generality of `GeoR` across diverse LLM backbones. The models differ in their base architectures, number of layers, hidden dimensions, and reserved special tokens (`<think>`, `</think>`) for explicit reasoning phase delineation during inference.

## A.7. Detailed Mathematical Derivation of Evaluation Metrics

In this appendix, we provide the formal definitions and detailed derivations of the evaluation metrics used in this paper, namely AUROC, AUPR, and FPR@95, for assessing the alignment between `GeoR` and the correctness of answers.

### A.7.1. PROBLEM SETUP AND NOTATIONS

We assume a dataset consisting of $B$ samples $\{(Q_i, \mathcal{E}_i, A_i^{\mathrm{GT}})\}_{i=1}^{B}$, where:

*Table 11.* Details of the downstream reasoning datasets (vertical-domain)

| Datasets (Train/Test) | GSM8K (7473/1319) | CommonsenseQA (9741/1140) | TheoremQA (-/800) |
|---|---|---|---|
| **Venue** | (Cobbe et al., 2021) | (Talmor et al., 2019) | (Chen et al., 2023) |
| **Task** | Math word problem solving | Commonsense reasoning | Mathematical theorem reasoning |
| **Metric** | Accuracy (f) | Accuracy (f) | Accuracy (f) |
| **Answer Format** | Number | Option | Option / Boolean / Number / List of Numbers |
| **Prompt Template** | Solve this math problem. Provide reasoning steps before giving the final answer in the format "Answer:". Only output the integer after "Answer:". 

**Question:** {input_data} | Answer the multiple choice commonsense question. Final line format: "Answer: $LETTER" (A/B/C/D/E). Provide step-by-step reasoning. 

**Question:** {input_data} | Complete the math problem considering the provided answer type. Different formats are required for boolean, integer, list of numbers, or option. 

**Question:** {input_data} 
**Answer Type:** {answer_type} |
| **Example** | **Q**: Natalia sold clips to 48 friends in April, and half as many in May. How many in total? 
**A**: 72 | **Q**: The sanctions against the school were a punishing blow. They seemed to what the efforts the school made to change? Options: A. ignore B. enforce C. authoritarian D. yell at E. avoid 
**A**: A | **Q**: How many ways are there to divide a set of 8 elements into 5 non-empty ordered subsets? 
**A**: 11760 |

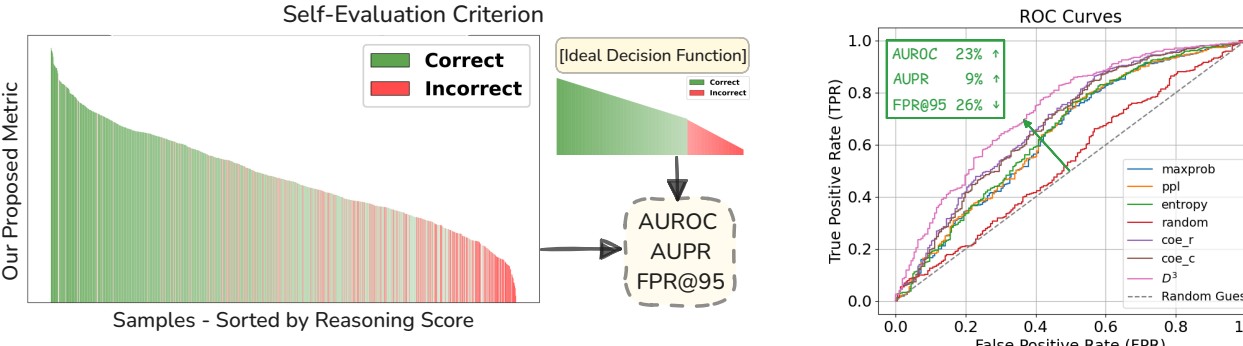

*Figure 7.* Left: Samples ranked by `GeoR` score, where higher scores align with correct outputs, reflecting the soundness of reasoning and logical causal relationship in terms of AUROC, AUPR, and FPR@95 metric. Right: ROC curves comparing `GeoR` score with baselines, where the dashed line denotes the expected performance of a random scoring function as a lower bound.

- $Q_i$ denotes the $i$-th query.

- $\mathcal{E}_i$ denotes the `GeoR` computed by our method for sample $i$.

- $A_i^{\mathrm{GT}} \in \{0, 1\}$ is the ground-truth correctness label (1 for correct, 0 for incorrect).

We further define:

- $\mathcal{P} = \{i \mid A_i^{\mathrm{GT}} = 1\}$ : set of indices for positive (correct) samples.

- $\mathcal{N} = \{j \mid A_j^{\mathrm{GT}} = 0\}$ : set of indices for negative (incorrect) samples.

### A.7.2. AUROC (AREA UNDER RECEIVER OPERATING CHARACTERISTIC)

AUROC measures the probability that a randomly chosen positive sample is ranked higher than a randomly chosen negative sample based on `GeoR`. It is formally computed as:

*Table 12.* Details of the multi-domain and multilingual evaluation datasets

| Datasets (Train/Test) | MMLU (99842/14042) | MMLU-Pro (-/12032) | MGSM (-/250 per language) |
|---|---|---|---|
| **Venue** | (Hendrycks et al., 2021) | (Wang et al., 2024a) | (Shi et al., 2023) |
| **Task** | Multitask multiple-choice QA | Robust multitask multiple-choice QA | Multilingual math word problem solving |
| **Metric** | Accuracy (f) | Accuracy (f) | Accuracy (f) |
| **Answer Format** | Option | Option | Number |
| **Prompt Template** | Answer the multiple choice question. Final line format: "Answer: $LETTER". **Question:** {input_data} | Same as MMLU. **Question:** {input_data} | Solve the math problem. Output final answer after "Answer:". **Question:** {input_data} |
| **Example** | **Q**: Let $p = (1, 2, 5, 4)(2, 3)$ in $S_5$. Find the index of $\langle p \rangle$ in $S_5$. Options: A. 8  B. 2  C. 24  D. 120 **A**: C | **Q**: Find the characteristic of the ring $2\mathbb{Z}$. Options:  A. 0  B. 30  C. 3  D. 10  E. 12  F. 50  G. 2  H. 100  I. 20  J. 5 **A**: A | **Q**: A robe takes 2 bolts of blue fiber and half that much white fiber. How many bolts in total? **A**: 3 |

$$\text{AUROC} = \frac{1}{|\mathcal{P}||\mathcal{N}|} \sum_{i \in \mathcal{P}} \sum_{j \in \mathcal{N}} \mathbb{K}(\mathcal{E}_i > \mathcal{E}_j), \tag{5}$$

where $\mathbb{K}(\cdot)$ is the indicator function:

$$\mathbb{K}(\mathcal{E}_i > \mathcal{E}_j) = \begin{cases} 1, & \text{if } \mathcal{E}_i > \mathcal{E}_j, \\ 0, & \text{otherwise.} \end{cases} \tag{6}$$

A higher AUROC (closer to 1.0) indicates better global ranking capability.

### A.7.3. AUPR (AREA UNDER PRECISION-RECALL CURVE)

AUPR quantifies the precision-recall trade-off, especially important under class imbalance. At a threshold $t$, precision and recall are defined as:

$$\text{Precision}(t) = \frac{TP(t)}{TP(t) + FP(t)}, \tag{7}$$

$$\text{Recall}(t) = \frac{TP(t)}{TP(t) + FN(t)}, \tag{8}$$

where $TP(t)$, $FP(t)$, and $FN(t)$ denote true positives, false positives, and false negatives at threshold $t$, respectively.

AUPR is computed as the area under the precision-recall curve:

$$\text{AUPR} = \int_0^1 \text{Precision}(\text{Recall}) \, d(\text{Recall}), \tag{9}$$

which is typically approximated using numerical integration methods (e.g., trapezoidal rule).

### A.7.4. FPR@95 (FALSE POSITIVE RATE AT 95% RECALL)

FPR@95 measures the false positive rate at the threshold where the recall reaches exactly $95\%$, evaluating robustness under stringent correctness requirements.

We first find the decision threshold $t^*$ such that:

$$t^* = \arg\min_t |\text{Recall}(t) - 0.95| . \tag{10}$$

*Table 13.* Model configuration details used in our experiments.

| Model Name | Base Model | Number of Layers | Hidden Size | Special Token IDs |
|---|---|---|---|---|
| DeepSeek-R1-Distill-Qwen-1.5B | Qwen2.5-Math-1.5B | 28 | 1536 | `<think>`: 151648 `</think>`: 151649 |
| DeepSeek-R1-Distill-Qwen-7B | Qwen2.5-Math-7B | 28 | 3584 | `<think>`: 151648 `</think>`: 151649 |
| DeepSeek-R1-Distill-Llama-8B | Llama-3.1-8B | 32 | 4096 | `<think>`: 128013 `</think>`: 128014 |
| DeepSeek-R1-Distill-Qwen-14B | Qwen2.5-14B | 48 | 5120 | `<think>`: 151648 `</think>`: 151649 |
| DeepSeek-R1-Distill-Qwen-32B | Qwen2.5-32B | 64 | 5120 | `<think>`: 151648 `</think>`: 151649 |
| DeepSeek-R1-Distill-Llama-70B | Llama-3.3-70B-Instruct | 80 | 8192 | `<think>`: 128013 `</think>`: 128014 |
| QwQ-32B | Qwen2.5-32B | 64 | 5120 | `<think>`: 151667 `</think>`: 151668 |

Then, the FPR@95 is computed as:

$$\text{FPR@95} = \frac{FP(t^*)}{|\mathcal{N}|}. \tag{11}$$

Lower FPR@95 indicates that `GeoR` reliably identifies correct reasoning trajectories with minimal false positives even at high-recall thresholds.

### A.8. Discussion: Physical Projections of `GeoR`

In developing `GeoR`, we observed that its formulation resonates with certain dynamical concepts in physics. While this connection should not be regarded as a strict proof, it offers an interesting perspective on why the components—distance, dynamism, and difficulty—combine effectively.

A.8.1. SETUP AND NOTATION

Let $\bar{q}^i \in \mathbb{R}^d$ denote the layer-averaged hidden state at the $i$-th layer ($i = 0, \ldots, L$). The geometric components $D_1$ (Distance) and $D_2$ (Dynamism) are defined as:

$$D_1 = \sum_{i=1}^{L} \|\Delta \bar{q}^i\|, \quad \text{where } \Delta \bar{q}^i = \bar{q}^i - \bar{q}^{i-1} \tag{12}$$

$$D_2 = \sum_{i=2}^{L} \|\Delta^2 \bar{q}^i\|, \quad \text{where } \Delta^2 \bar{q}^i = \Delta \bar{q}^i - \Delta \bar{q}^{i-1} \tag{13}$$

The difficulty correction term $\mathcal{H}$ (Calibration) is the average negative log-likelihood:

$$\mathcal{H} = \frac{1}{K} \sum_{k=1}^{K} [-\log p(q_k | q_{<k}, \mathbf{x})]. \tag{14}$$

The final score normalizes the geometric drive by the total displacement $\|\bar{q}^L - \bar{q}^0\|$ and subtracts the difficulty penalty:

$$\texttt{GeoR} = \frac{D_1 + D_2}{\|\bar{q}^L - \bar{q}^0\|} - \mathcal{H}. \tag{15}$$

A.8.2. PROJECTION I: KINEMATICS AND NORMALIZED EFFICIENCY

**Distance as Path Length.** $D_1$ represents the total arc-length of the reasoning trajectory in the hidden state space. In kinematics, a long path relative to displacement (high tortuosity) often implies inefficiency. However, in reasoning, we find that *normalized* path length ($D_1/\|\bar{q}^L - \bar{q}^0\|$) indicates "depth of thought"—the ratio of effective reasoning steps to the simple origin-to-destination vector. A higher ratio suggests the model is engaging in necessary complex transformations rather than taking a shortcut.

**Dynamism as Acceleration.** $D_2$ aggregates second-order changes, analogous to acceleration or curvature. Physically, this corresponds to the force applied to steer the reasoning process. A trajectory with high normalized dynamism implies the model is actively correcting and refining its internal state, rather than drifting passively.

A.8.3. PROJECTION II: ENERGETICS AND COST POTENTIALS

**Difficulty as an Energy Barrier.** The term $\mathcal{H}$ can be viewed as an *uncertainty potential* or energy barrier. High entropy ($\mathcal{H}$) indicates that the model is traversing a "high-energy" region of the solution space where the probability mass is diffuse (i.e., the model is unsure).

**The "Work minus Friction" Analogy.** The structure of `GeoR` mirrors a balance between *useful work* and *informational friction*:

$$\texttt{GeoR} \approx \text{Geometric Work} - \text{Uncertainty Penalty} \tag{16}$$

The normalized geometric terms represent the "work" done by the model to evolve the hidden states from input to answer. The entropy term $\mathcal{H}$ represents the "friction" or resistance encountered due to ambiguity. A valid reasoning path exerts sufficient work (high geometric evolution) to overcome the friction of uncertainty. If the work is high but uncertainty is also high (e.g., hallucination), the score cancels out. If the work is low (shallow reasoning), the score remains low. The highest scores are achieved when the model performs significant geometric work with minimal uncertainty.

A.8.4. SUMMARY OF PHYSICAL CORRESPONDENCE

- **Normalized Distance ($D_1$ terms):** Corresponds to efficient traversal (depth of thought).

- **Dynamism ($D_2$ terms):** Corresponds to steering forces (adaptive reasoning).

- **Calibration ($\mathcal{H}$):** Corresponds to potential energy barriers (uncertainty).

This physical lens clarifies why these specific components—one spatial, one temporal/derivative-based, and one probabilistic—form a complementary set for evaluating reasoning fidelity.

### A.9. Prompts Used per Dataset

Below we enumerate the exact prompts we used for each dataset during inference. For clarity, each prompt is shown inside a blue framed box.

---

**GSM8K Prompt**

Solve this math problem. Give the reasoning steps before giving the final answer on the last line by itself in the format of "Answer:".
Do not add anything other than the integer answer after "Answer:".
Question:
{input_data}

---

---

## mgsm Prompt

Solve this math problem. Giving the final answer by itself in the format of "Answer:".
Do not add anything other than the integer answer after "Answer:".
Question:
`{input_data}`

---

## mgsm_fr Prompt

Résolvez ce problème de mathématiques. Donnez la réponse finale sur la dernière ligne elle-même dans le format de "Réponse:".
N'ajoutez rien d'autre que la réponse entière après "Réponse:".
Question:
`{input_data}`

---

## mgsm_de Prompt

Löse dieses Mathematikproblem. Gib die endgültige Antwort in der letzten Zeile alleine im Format "Antwort:" an.
Füge nichts anderes als die ganzzahlige Antwort nach "Antwort:" hinzu.
Frage:
`{input_data}`

---

## mmlu Prompt

Answer the following multiple choice question. The last line of your response should be of the following format: "Answer: LETTER" (without quotes) where LETTER is one of A/B/C/D.
Question:
`{input_data}`

---

## CommonsenseQA Prompt

Answer the following multiple choice common-sense reasoning question. The last line of your response should be of the following format: "Answer: LETTER" (without quotes) where LETTER is one of A/B/C/D/E. Think step by step and output the reasoning process before answering.
`{input_data}`

---

## TheoremQA Prompt

Below is an instruction that describes a task, paired with an input that provides further context.
Write a response that appropriately completes the request.

**Instruction:** Read a math problem. If Answer Type is [bool], respond True/False. If numeric, respond accordingly. If list, respond like [2,3,4]. If option, respond like (a)/(b)/(c)/(d). Use final sentence: "Therefore, the answer is ...".

Question: `{input_data}`
Answer_type: `{answer_type}`
Response:

