# OpenReview forum: "The Geometry of Reasoning: Self-Evaluation via Layerwise Trajectory Evolution"
_ICML.cc/2026/Conference — ICML 2026 regular_

### Official Review · Reviewer_TGxp · 2026-03-11

**Soundness:** 3
**Presentation:** 2
**Significance:** 2
**Originality:** 3
**Overall Recommendation:** 4
**Confidence:** 3

**Summary:**

This paper introduces GeoR, a self-evaluation framework that uses internal model signals to verify the correctness of Chain-of-Thought (CoT) outputs in Large Reasoning Models (LRMs). It primarily uses the model's internal hidden states to measure how closely the generated CoT aligns with the correct one. Additionally, it incorporates a difficulty-aware calibration mechanism that relies on the cross-entropy of the reasoning trajectory to gauge both problem difficulty and generation uncertainty. GeoR has been tested on multiple models and datasets, outperforming various self-evaluation baselines. The paper also includes supplementary analyses on computational latency, open-world distribution shifts, and ablation studies on key formula components and pooling strategies.

**Compliance With Llm Reviewing Policy:**

Affirmed.

**Final Justification:**

Fully resolved - My concerns have been adequately addressed.

**Key Questions For Authors:**

1.GeoR uses global average pooling for all reasoning tokens in each layer. Have you analyzed if this causes information loss and how significant the impact is?
2.Are there any plans to run related experiments on MoE-based LLMs?
3.When a model gets stuck in repetition or generates long-winded nonsense, the generation probability is usually high and the layer evolution looks seemingly stable. Can GeoR effectively identify these abnormal trajectories?

**Limitations:**

yes

**Strengths And Weaknesses:**

Strengths:
1.The authors propose measuring the internal reasoning quality of large reasoning models through the geometric variations of layer-wise hidden states, offering a novel perspective for white-box self-evaluation.
2.The GeoR framework is computationally efficient, requiring a single forward pass to extract features and complete computations.
3.The experiments are comprehensive, with evaluations conducted across different parameter scales ranging from 1.5B to 70B, various model architectures, and multimodal scenarios.
Weaknesses:
1.The framework applies global average pooling to the hidden states of all reasoning tokens at each layer, which may lead to information loss. This potential issue lacks thorough analysis.
2.The evaluation does not include the Mixture of Experts (MoE) architecture, which is currently a mainstream model structure.
3.There is insufficient discussion on extreme or abnormal reasoning trajectories. The paper lacks a detailed analysis of GeoR's performance when large models fall into circular reasoning (the repetition/hallucination loop phenomenon) or generate overly long, irrelevant trajectories.

---

> ### Author Rebuttal · Authors · 2026-03-30
>
> We sincerely thank the reviewer for recognizing GeoR's **novel white-box perspective** and **computational efficiency**.
>
> > **Weakness 1 & Question 1: Have you analyzed if global average pooling causes information loss and how significant the impact is?**
>
> Yes. We use **mean pooling intentionally**, not for convenience but for adopting the most appropriate way to obtain a sequence-level reasoning representation for self-evaluation. The key reasons are as follows.
>
> 1. **It is a standard way to extract global semantics.**
>
> Mean pooling has been widely adopted in semantic representation learning (e.g., Sentence-BERT[1], SimCSE[2]), and we follow the same principle here: aggregating token-level states into a stable sequence-level summary. In our setting, the pooled object is not the full output, but specifically the **reasoning segment**, so the goal is to preserve the shared reasoning semantics while suppressing token-level idiosyncrasies and local lexical noise. This design choice is also stated explicitly in our method section.
>
> 2. **It is effective because self-evaluation needs the global reasoning state, not isolated token details.**
>
> GeoR models **layer-wise evolution** of the reasoning process. For that purpose, a stable summary of the whole reasoning segment is more suitable than raw token-wise dynamics. The latter way with local word changes would easily dominate rather than reflecting the genuine reasoning progress. Our ablation also directly proves this: mean reasoning consistently outperforms alternative constructions, improving AUROC by about **3–7 points** over last-token or global-output pooling and substantially reducing FPR@95 across models.
>
> 3. **Without mean pooling, the representation becomes less faithful to reasoning quality.**
>
> This is the key reason we regard mean pooling as the preferred choice. If we use only the **last CoT token**, we lose most of the trajectory history; if we average **all output tokens**, the signal is diluted by conclusion / non-reasoning tokens; if we track **token-wise dynamics**, performance drops sharply because the representation is driven by lexical variation instead of depth-wise reasoning evolution. Table 2 shows exactly this pattern: mean reasoning is consistently the strongest option, while token-wise dynamics are often near random. In this sense, mean pooling is not merely acceptable; it is the **best compromise between information retention and noise suppression**.
>
> [1] Sentence-BERT: Sentence Embeddings using Siamese BERT-Networks. EMNLP 2019.
>
> [2] SimCSE: Simple Contrastive Learning of Sentence Embeddings. EMNLP 2021.
>
> > **Weakness 2 & Question 2: Are there any plans to run related experiments on MoE-based LLMs?**
>
> Yes. We have **already added** an additional experiment on the **MoE-based model Mixtral-8x7B**, evaluated on **GSM8K**. Our method captures universal properties of reasoning trajectories rather than dense-transformer artifacts. As shown below, although sparse routing alters the exact scale of hidden-state transitions, the core qualitative trend and effectiveness metric of GeoR remain highly consistent.
>
> | Method                |    AUROC ↑    |     AUPR ↑     |    FPR@95 ↓    |
> | :-------------------- | :-------------: | :-------------: | :-------------: |
> | Random                |      50.01      |      46.51      |      92.91      |
> | CoE-R                 |      61.11      |      53.82      |      86.15      |
> | **GeoR (ours)** | **68.47** | **65.31** | **83.86** |
>
> > **Weakness 3 & Question 3: Can GeoR effectively identify abnormal trajectories such as repetition or hallucination loops?**
>
> **Yes, GeoR can identify the mentioned trajectories.** To prove it, we added a targeted stress test where we isolated a subset comprising the top 10% longest reasoning responses from the MMLU-Pro dataset (evaluated on R1-LLaMA-8B), a subset predominantly characterizing extended or potentially looping logic.
>
> | Method                |    AUROC ↑    |     AUPR ↑     |    FPR@95 ↓    |
> | :-------------------- | :-------------: | :-------------: | :-------------: |
> | Random                |      47.99      |      13.08      |      93.61      |
> | CoE-R                 |      48.24      |      11.19      |      94.25      |
> | **GeoR (ours)** | **57.40** | **14.92** | **88.18** |
>
> As demonstrated above, GeoR consistently maintains its evaluation efficacy and decisively outperforms standard baselines even under extreme redundancy conditions. This further confirms that our framework successfully filters out the interference caused by long-winded, hallucinated loops.
>
> We hope these clarifications thoroughly resolve your concerns and positively influence your final evaluation.

---

> > ### Author Rebuttal · Reviewer_TGxp · 2026-04-04
> >
> > Thanks for detailed response. I have increased score.

---

> > > ### Author Response · Authors · 2026-04-07
> > >
> > > Thank you very much for the thoughtful follow-up and for confirming that your concerns have been fully resolved. We are particularly encouraged that our rebuttal helped clarify the key issues and led you to raise your score. Your strengthened positive assessment is highly valuable to us, and we sincerely appreciate your careful evaluation, constructive engagement, and generous support for our work.

---

### Official Review · Reviewer_yBh8 · 2026-03-12

**Soundness:** 4
**Presentation:** 4
**Significance:** 3
**Originality:** 3
**Overall Recommendation:** 5
**Confidence:** 4

**Summary:**

This paper introduces **GeoR (Geometry of Reasoning)**, a novel white-box self-evaluation framework designed for Large Reasoning Models (LRMs). The authors address the challenge of enabling LRMs to assess the correctness of their own generated Chain-of-Thought (CoT) trajectories without external supervision or ground-truth labels.
GeoR's core contribution lies in its shift from surface-level output probabilities to the **layerwise evolution** of hidden-state trajectories. The framework consists of two primary components:



1. **Geometric Evolution:** Quantifies reasoning progress by synthesizing first-order variations ($D_1$, representing path length) and second-order variations ($D_2$, representing trajectory curvature/adaptability) of hidden states across transformer layers.





2. **Difficulty-Aware Calibration:** Utilizes the average negative log-likelihood (cross-entropy) of the reasoning tokens to normalize the geometric signal against the intrinsic uncertainty of the query.





The authors demonstrate through extensive experiments on eight models (ranging from 1.5B to 70B parameters) and seven diverse benchmarks that GeoR consistently outperforms state-of-the-art white-box and black-box baselines in metrics such as AUROC, AUPR, and FPR@95. Notably, the method adds negligible computational overhead ($1.2 \times 10^{-3}s$), making it highly efficient for real-world deployment compared to sampling-based alternatives.

**Compliance With Llm Reviewing Policy:**

Affirmed.

**Key Questions For Authors:**

1. **Generalization to Non-Reasoning Models:** While GeoR is designed for LRMs with explicit `<think>` trajectories, could the geometric evolution of hidden states still serve as a confidence signal for standard LLMs during the generation of the final answer?

2. **Layer Contributions:** Have you experimented with using only a subset of layers? For instance, does the "geometric work" in early layers (often associated with syntactic processing) correlate with correctness as strongly as the "work" in deep layers?

3. **Handling Hidden Trajectories:** Some models now hide their reasoning tokens from the output but still process them internally. Does GeoR require the *explicit* generation of tokens, or can it function on the hidden states of tokens that are processed but not sampled?

**Limitations:**

Yes

**Strengths And Weaknesses:**

### Strengths

**Soundness:** The paper provides a very thorough empirical validation. Testing across a vast array of model scales (1.5B to 70B), training paradigms (SFT, RL, Distillation), and modalities (including vision-language via Qwen2.5-VL) suggests that the findings are robust and not merely artifacts of specific architectures.

**Originality:** The insight to model "reasoning progression" along the **network depth (layers)** rather than the **temporal sequence (tokens)** is a creative and well-justified pivot. This effectively treats the transformer's layers as a refinement process for the model's "depth of thought".

**Significance:** Reliability and self-correction are critical hurdles for LRMs. GeoR offers a "near-zero cost" solution for real-time monitoring. Its ability to maintain performance under open-world distributional shifts is highly relevant for practical AI safety.

**Presentation:** The manuscript is exceptionally well-structured. The mathematical formulations are clear (Section 2.2), and the visualizations (especially Figure 2 for intuition and Figure 4 for holistic comparison) effectively communicate the core results.

### Weaknesses

**Access Requirements:** As a white-box method, GeoR requires access to internal hidden states and token-level probabilities. This limits its applicability to open-weight models, making it unusable for closed-source, API-only models (e.g., OpenAI o1-preview) that are currently leading the LRM field.

**Theoretical Rigor of "Physics Projections":** While the discussion in Appendix A.8 regarding kinematics and energetics is an interesting mental model, it remains a "projection" rather than a rigorous theoretical proof. The connection between $L_2$ norms in high-dimensional latent space and physical work is more metaphorical than mathematical.

**Layer Sensitivity:** The paper uses all layers for the calculation. It would be beneficial to know if specific subsets of layers (e.g., mid-to-late layers where semantics are typically refined) contribute more significantly to the signal, which could further optimize the computation.

---

> ### Author Rebuttal · Authors · 2026-03-30
>
> ## Response
>
> We sincerely thank the reviewer for recognizing GeoR's **originality** and **strong empirical validation**.
>
> > **Weakness 1: As a white-box method, GeoR requires access to internal hidden states and token-level probabilities.**
>
> GeoR is a **white-box method by design**, which is also the main presumption of this paper. More broadly, most strong self-evaluation methods in this line of work rely on **white-box access**, while purely black-box alternatives usually provide weaker signals and require much higher cost (e.g., repeated sampling or prompting).
>
> > **Weakness 2: The physical analogy in Appendix A.8 remains a "projection" rather than a rigorous theoretical proof.**
>
> We agree that the physical analogy is **not** intended as a formal theoretical foundation for the method. It is included purely as an **intuitive metaphor** to help readers’ understanding. The analogy is not required for either the formulation or the empirical results.
>
> Deriving a rigorous proof linking hidden-state geometry to cognitive correctness remains a notoriously elusive open problem in deep learning. Since modern LRMs exhibit highly opaque internal computations—where even explicit CoT tokens often unfaithfully reflect actual cognition [1,2]—sample-level theoretical guarantees are currently unachievable. Consequently, our contribution establishes a principled, statistically robust empirical proxy across massive distributions rather than proposing an unprovable universal theorem. For further details, please refer to our response to Reviewer CvFZ (Weakness 1 & Question 1).
>
> [1] *Let’s Verify Step by Step*. OpenAI, 2023.
>
> [2] *Language Models Don't Always Say What They Think*. Anthropic, 2023.
>
> > **Weakness 3 & Question 2: Have you experimented with using only a subset of layers?**
>
> We deeply appreciate this insightful perspective. To investigate layer sensitivity, we ablated the geometric evolution across specific depth spans on the MMLU-Pro dataset utilizing R1-LLaMA-8B:
>
> | Layer Span Configuration | AUROC ↑ | AUPR ↑ | FPR@95 ↓ |
> | :--- | :--------: | :--------: | :----: |
> | Random | 50.95 | 31.25 | 94.30 |
> | **Early Syntactic Layers (0-15)** | 54.91 | 34.24 | 91.74 |
> | **Deep Semantic Layers (16-32)** | 58.36 | 36.81 | 90.11 |
> | **Full Cognitive Depth (0-32)** | **60.47** | **37.90** | **89.24** |
>
> **Observation:** Early layers primarily process syntactic formulation and exhibit weaker correlations with reasoning correctness. In contrast, deep layers, which execute semantic refinement and logical deduction, capture a substantially stronger geometric signal. Nevertheless, integrating the full trajectory remains strictly optimal, as it comprehensively encapsulates the unbroken end-to-end cognitive evolution.
>
> > **Question 1: Could the geometric evolution of hidden states still serve as a confidence signal for standard LLMs during the generation of the final answer?**
>
> **Yes, GeoR works seamlessly without explicit CoT traces.** We utilized `<think>` tags in our main experiments simply because doing so aligns with the standard conventions of current cutting-edge LLMs. However, special formatting tokens are by no means a prerequisite for our method.
>
> Applying GeoR to standard models without special tags is plug-and-play. For instance, when evaluating Qwen2-7B on GSM8K, we naturally isolated the reasoning trajectory using a standard rule-based parsing function—a default and widely-adopted practice for final answer extraction.
>
> | Method         |         AUROC↑ |          AUPR↑ |         FPR95↓ |
> | :------------- | --------------: | --------------: | --------------: |
> | Random         |           49.75 |           68.25 |           91.30 |
> | CoE            |           68.31 |           73.82 |           84.25 |
> | **GeoR** | **75.47** | **81.20** | **78.24** |
>
> These results confirm that GeoR works very well with or without explicit system tokens.
>
> > **Question 3: Does GeoR require the explicit generation of tokens, or can it function on the hidden states of tokens that are processed but not sampled?**
>
> Yes, GeoR **does not require explicitly generated tokens** and can operate on hidden reasoning states. GeoR's geometric evaluation intrinsically tracks latent representation trajectories independently of the visible textual output. Consequently, it naturally extends to hidden or un-sampled reasoning states as long as those internal states are accessible. For such scenarios, only our entropy-based calibration term ($H$) requires redefinition—transitioning from token-level perplexity to a continuous latent uncertainty metric. We leave this adaptation to future exploration, given that explicitly emitted CoT reasoning remains the prevailing paradigm for open-weight LRMs.
>
> We hope these clarifications will further strengthen your already positive evaluation.

---

### Official Review · Reviewer_ENMi · 2026-03-14

**Soundness:** 3
**Presentation:** 3
**Significance:** 3
**Originality:** 3
**Overall Recommendation:** 4
**Confidence:** 3

**Summary:**

This paper introduces GeoR, a white-box self-evaluation metric for Large Reasoning Models (LRMs). Instead of relying on expensive sampling (like self-consistency) or external LLM judges, GeoR assesses correctness by analyzing the geometric trajectory of internal hidden states across layers. The core scoring function combines first-order progress ($D_1$) and second-order adaptability ($D_2$), normalized by the total displacement $\| \bar{q}^L - \bar{q}^0 \|$, and calibrated with an entropy penalty ($H$). The authors test this across 8 models (from 1.5B to 70B) and 7 benchmarks, demonstrating SOTA performance for confidence estimation with practically zero latency overhead (roughly 1.2ms).

**Compliance With Llm Reviewing Policy:**

Affirmed.

**Key Questions For Authors:**

1. Can you defend the specific formulation of $D_1 + D_2 - H$ against other geometric properties? Did you test alternatives like trajectory curvature or geodesic distance on the representation manifold, and did they fail?

2. How exactly are you segmenting "reasoning tokens" in models that do not use explicit <think> or <scratchpad> delimiters? Does the performance tank if this boundary is fuzzy?

**Limitations:**

The authors are upfront about the white-box requirement, but they need to address the assumption of identifiable reasoning tokens and the lack of theoretical guarantees. The paper would also benefit from a brief discussion on how this method might fare on fundamentally different architectures (e.g., MoE or state-space models) where representation dynamics might look entirely different.

**Strengths And Weaknesses:**

1. The core intuition—that correct reasoning exhibits a smooth, progressive geometric trajectory while hallucinations or logical errors "thrash around" in the latent space—is highly intuitive. Translating this into a measurable geometric formulation is a very neat conceptual leap.

2. Testing across 8 different model scales, and specifically including multimodal (Geometry3K) and multilingual (MGSM) setups, makes the empirical claims highly credible and proves the metric isn't just overfitting to standard English text.

3. Coming in at $1.2 \times 10^{-3}$ seconds, GeoR is orders of magnitude cheaper than self-consistency or prompt-based self-reflection. This makes it a highly viable tool for real-time deployment and inference-time compute allocation.

but

1. The Mean Reasoning pooling strategy relies heavily on isolating reasoning tokens from the rest of the prompt/output. This is trivial for models that output explicit <think> tags, but it's highly ambiguous for standard LLMs. The paper doesn't clearly explain how these boundaries are defined in the wild.

---

> ### Author Rebuttal · Authors · 2026-03-30
>
> ## Response
>
> We sincerely thank the reviewer for recognizing GeoR's **core intuition** and **extreme efficiency**.
>
> > **Weakness & Question 2: How exactly are you segmenting "reasoning tokens" in models that do not use explicit delimiters?**
>
> **Yes, GeoR works seamlessly without explicit CoT traces.** We utilized `<think>` tags in our main experiments simply because doing so aligns with the standard conventions of current cutting-edge reasoning models (LRMs), making extraction convenient. However, special formatting tokens are by no means a prerequisite for our method. Please refer to our response to Reviewer KRN6 (Question 1) for detailed empirical results on implicit CoT settings.
>
> > **Question 1: Can you defend the specific formulation of GeoR against other geometric properties?**
>
> Rather than claiming ours is the unique geometric statistic, we propose it as a **minimal, interpretable decomposition** of layerwise evolution. The reviewer’s suggested "curvature" is intrinsically captured by our second-order term $D_2$, designed precisely to model adaptive refinement rather than mere path length.
>
> We selected this specific formulation, $\frac{D_1 + D_2}{\|\Delta q\|}$, because it simultaneously satisfies three desiderata: capturing both forward progress ($D_1$) and adaptive correction ($D_2$), remaining strictly **parameter-free and single-pass**, and enabling cross-architecture comparability via displacement normalization.
>
> While alternative manifold-geodesic metrics exist, they impose stronger assumptions on the underlying metric structure and often introduce nontrivial hyperparameters. More broadly, other geometric constructions are certainly possible, but they may **be more tightly bound to specific model architectures or particular reasoning mechanisms**. In contrast, our formulation is intentionally more **generalized**, capturing the major first- and second-order geometric characteristics in a lightweight and architecture-agnostic way. Exploring richer geometric variants is an interesting direction for future work.
>
> > **Limitation: Lack of Theoretical Guarantees**
>
> Frankly speaking, establishing a strictly necessary condition for reasoning fidelity is **a common ultimate goal of the research community, and thus remains an open challenge** [1,2]. Given the profound architectural complexity of LLMs as well as the inherent weak interpretability of their internal computational dynamics, we acknowledge that no existing protocol can theoretically rule out all potential "unmet conditions" or counter-examples. For a deeper discussion, please refer to our response to Reviewer CvFZ (Weakness 1 & Question 1).
>
> [1] Let’s Verify Step by Step. OpenAI
> [2] Language Models Don't Always Say What They Think. Anthropic
>
> > **Limitation: Applicability to Fundamentally Different Architectures**
>
> Yes. We added an additional experiment on the **MoE-based model Mixtral-8x7B**, evaluated on **GSM8K**. Our method captures universal properties of reasoning trajectories rather than dense-transformer artifacts. As shown below, although sparse routing alters the exact scale of hidden-state transitions, the core qualitative trend and effectiveness metric of GeoR remain highly consistent.
>
> | Method | AUROC ↑ | AUPR ↑ | FPR@95 ↓ |
> | :-- | :--: | :--: | :--: |
> | Random | 50.01 | 46.51 | 92.91 |
> | CoE-R | 61.11 | 53.82 | 86.15 |
> | **GeoR (ours)** | **68.47** | **65.31** | **83.86** |
>
> Since SSMs do not yet constitute the primary paradigm for large reasoning models, we plan to explore their state-space geometric evolution in future work and update the revised manuscript accordingly.
>
> We hope these clarifications will further strengthen your positive evaluation.

---

> > ### Author Rebuttal · Reviewer_ENMi · 2026-04-04
> >
> > Thank you for the responsive rebuttal.
> >
> > **Reasoning token segmentation (Weakness & Q2):** The clarification that GeoR does not require explicit `<think>` delimiters, along with the pointer to empirical results on implicit CoT settings, addresses my main concern. The practical applicability is broader than I initially assumed.
> >
> > **Formulation defense (Q1):** The explanation that the second-order term already captures curvature-like behavior is reasonable. The three desiderata (forward progress + adaptive correction, parameter-free single-pass, displacement normalization for cross-architecture comparability) provide a clear rationale for the specific design choices over alternative geometric constructions.
> >
> > **MoE architecture (Limitation):** The additional Mixtral-8x7B experiment on GSM8K is a welcome addition and demonstrates that GeoR generalizes beyond dense transformers, which was a gap I flagged. The consistent trend across sparse routing validates the architecture-agnostic claim.
> >
> > All concerns from my original review have been adequately addressed. I keep my original score.

---

> > > ### Author Response · Authors · 2026-04-07
> > >
> > > Thank you very much for the thoughtful follow-up and for confirming that all concerns from your original review have been adequately addressed. We are particularly encouraged by your recognition of GeoR’s broader practical applicability, the rationale behind its geometric formulation, and its generalization beyond dense transformers. Your clear and supportive assessment is greatly appreciated, and we sincerely thank you for your careful reading, constructive feedback, and positive evaluation of our work.

---

### Official Review · Reviewer_CvFZ · 2026-03-20

**Soundness:** 3
**Presentation:** 3
**Significance:** 3
**Originality:** 3
**Overall Recommendation:** 4
**Confidence:** 5

**Summary:**

This paper introduces GeoR (Geometry of Reasoning), a white-box self-evaluation framework for Large Reasoning Models (LRMs). The key insight is that layerwise trajectory evolution of hidden states provides a signal for assessing reasoning quality without external supervision. The method decomposes reasoning fidelity into two dimensions: (1) Geometric Evolution, which synthesizes first- and second-order hidden-state variations across transformer layers, and (2) Difficulty-Aware Calibration, which uses cross-entropy to normalize scores against query uncertainty. The authors evaluate GeoR across 8 LRMs (1.5B to 70B parameters) and 7 benchmarks, demonstrating consistent improvements over baselines in AUROC, AUPR, and FPR@95 metrics.

**Compliance With Llm Reviewing Policy:**

Affirmed.

**Final Justification:**

The authors answered the questions clearly, and I am more confident about my evaluation.

**Key Questions For Authors:**

1. Can you provide any theoretical justification or intuition for why geometric evolution (specifically D1 + D2 normalized by displacement) should correlate with reasoning correctness? The empirical correlation is shown, but what is the underlying mechanism?

2. The method assumes access to layer-wise hidden states and special reasoning tokens (<think>, </think>). How does this limit applicability to models that (a) don't use explicit reasoning tokens, or (b) are accessed only through APIs that don't expose hidden states? Could the method be adapted for black-box scenarios?

3. In Table 3, the ablation shows that using H (difficulty) alone performs worse than D1 or D2 alone (69.71 vs 74.35/74.07 AUROC). Yet the full method includes H as a subtraction term. Can you explain why penalizing high-entropy trajectories helps when H alone is not a good signal?

4. How does GeoR behave on adversarial or out-of-distribution inputs where the model might generate high-entropy but geometrically stable trajectories that are nonetheless incorrect?

**Limitations:**

The authors have adequately discussed technical limitations in Section 3.4 (ablation studies, sensitivity analysis) and acknowledge computational costs. The paper appropriately scopes its claims to reasoning tasks with explicit Chain-of-Thought trajectories, which is a reasonable and well-justified boundary.

**Strengths And Weaknesses:**

# Strengths:
- The specific combination of layer-wise geometric evolution with difficulty-aware calibration appears novel. The use of first- and second-order hidden-state variations to characterize reasoning quality is a creative approach.
- The insight that layer-wise dynamics (depth) are more informative than token-wise dynamics (time) for reasoning quality is an interesting contribution that challenges the typical temporal framing of CoT analysis.
- The theoretical formulation of Geometric Evolution (D1, D2) is well-grounded in the mechanics of transformer inference. The layer-wise modeling approach (Equation 1-4) is technically sound and aligns with how hidden states evolve through network depth.
- The experimental evaluation is extensive, covering diverse model families (Qwen, Llama), training paradigms (SFT, RL/GRPO, distillation), and task domains (math, commonsense, theorem proving, multilingual, multimodal).
- The ablation study (Table 2, Table 3) provides good evidence for design choices, particularly demonstrating why layer-wise evolution outperforms token-wise dynamics and why the normalization term is necessary.
- The statistical robustness analysis (Table 4) with multiple runs and standard deviations adds credibility to the claims.
- Self-evaluation without external supervision is a critical problem for LRM deployment, especially in safety-critical applications. The paper addresses this directly.
- The computational efficiency of GeoR (1.2ms overhead vs. 15.24s inference) makes it practical for real-world use, unlike sampling-based methods.
- The cross-lingual and multimodal generalization results (Table 7, Table 10) suggest the method captures language-agnostic properties of reasoning quality.

# Weaknesses
- The paper lacks theoretical analysis of why geometric evolution correlates with correctness. While empirical correlations are shown, there is no formal justification or provable guarantee.
- The comparison against baselines could be more comprehensive. While the paper cites semantic entropy methods (Kuhn et al., 2023; Farquhar et al., 2024) in the related work, these are not included in the empirical comparison. Semantic entropy represents a major recent approach to uncertainty quantification for LLMs and would provide a strong baseline for comparison.
- The physical analogy in Appendix A.8, while interesting, is somewhat hand-wavy and not formally connected to the method's performance.
- The broader idea of using hidden-state geometry for uncertainty estimation has precedents (e.g., Azaria & Mitchell 2023, Burns et al. 2024 are cited). The novelty lies in the specific formulation rather than the general concept.
- The Difficulty-Aware Calibration using cross-entropy is similar to existing perplexity-based methods, and the paper could better articulate what distinguishes H from standard perplexity measures.

---

> ### Author Rebuttal · Authors · 2026-03-30
>
> > **WK1 & Q1: Theoretical justification for geometric evolution?**
>
> Frankly speaking, establishing a strictly necessary condition for reasoning fidelity is a common ultimate goal of the research community, and thus remains an open challenge [1,2]. The whole community including us acknowledge that given the profound architectural complexity of LLMs, the interpretability of their internal computational dynamics is enormously weak. Therefore, no existing protocol can theoretically rule out all potential "unmet conditions" or counter-examples.
>
> Precisely due to the inherent weak interpretability, existing methods—including ours—currently rely on large-scale empirical experiments to validate the effectiveness of CoT evaluation metrics. Importantly, our exhaustive experiments demonstrate that among all existing characterization metrics, GeoR offers the highest accuracy.
>
> The core intuition behind GeoR is that high-fidelity reasoning very likely manifests as a structured, purposeful trajectory of hidden states across depth, rather than chaotic exploration. Specifically:
> *   $D_1$ captures the cumulative forward latent progress.
> *   $D_2$ measures adaptive structural correction.
> *   **Normalization** by end-to-end displacement eliminates raw scale bias, isolating true structured refinement from mere activation magnitude.
> *   **Subtracting $H$** fundamentally penalizes cases where large latent motion is driven by high-uncertainty hallucination rather than confident deduction.
>
> Collectively, GeoR mathematically characterizes  "purposeful internal evolution." As consistently corroborated by our extensive empirical ablations, correct reasoning exhibits stronger, more stable latent progress than incorrect or unfaithful traces—a ubiquitous pattern that GeoR successfully captures.
>
> Importantly, we do not claim this decomposition is uniquely correct in a theoretical sense. Rather, we offer a simple, interpretable, and parameter-free formulation that works best **so far** among the alternatives we evaluated, as supported by our ablations and large-scale empirical results.
>
> [1] Let’s Verify Step by Step. OpenAI, 2023.
> [2] Language Models Don't Always Say What They Think: Unfaithful Explanations in Chain-of-Thought Prompting. Anthropic, 2023.
>
> > **WK2: Comprehensive baseline comparisons.**
>
> GeoR remains clearly stronger than semantic entropy while being much more efficient. Semantic entropy requires multiple sampled generations and semantic clustering, which introduces substantial overhead. GeoR instead works in a single pass on internal representations. As shown below on GSM8K, GeoR still outperforms semantic entropy by a large margin on all metrics.
>
> M|AUROC↑|AUPR↑|FPR95↓
> -|-|-|-
> **GeoR**|**77.28**|**91.42**|**72.20**
> Semantic Entropy|59.47|85.51|83.65
>
> > **WK3: Physical analogy in Appendix**
>
> We agree that the physical analogy is **not** intended as a formal theoretical foundation for the method. It is included purely as an **intuitive metaphor** to help readers’ understanding. The analogy is not required for either the formulation or the empirical results.
>
> > **WK4: Novelty is mainly in the formulation.**
>
> Due to space limits, please see our comparison with prior geometry-based methods in Reviewer KRN6 (W2).
>
> > **WK5: Distinction of H from standard perplexity.**
>
> We agree $H$ is related to log-perplexity. Rather than treating it as a novel confidence score, GeoR uses $H$ solely as a lightweight penalty to calibrate the geometric signal, distinguishing purposeful internal refinement from high-uncertainty wandering.
> > **Q2: Adaptation to black-box scenarios?**
>
> For (a) regarding implicit reasoning tokens, please refer to our detailed response to Reviewer KRN6 (Question 1), where we demonstrate that GeoR seamlessly adapts to them. For **(b)**, GeoR is a **white-box method by design**, which is also the presumption of this paper. More broadly, most strong self-evaluation methods in this line of work rely on **white-box access**, while purely black-box alternatives usually provide weaker signals and require much higher cost (e.g., repeated sampling or prompting).
>
> > **Q3: Explain the role of entropy penalty (H).**
>
> $H$ is not intended to be a standalone correctness predictor. Instead, it serves as a crucial calibration term that penalizes high-uncertainty wandering among samples with similar geometric evolution. This explains why $H$ alone is weaker, yet consistently improves the full method by filtering out chaotic exploration once the primary geometric signal ($D_1 + D_2$) provides a structural foundation.
>
> > **Q4: Behavior on adversarial/OOD inputs.**
>
> GeoR's robustness to OOD inputs is shown in our open-world evaluation (Figure 4, Table 5). Furthermore, the adversarial scenario you query—geometrically stable trajectories under high uncertainty—is exactly the failure mode our Difficulty-Aware Calibration ($H$) selectively penalizes.
>
> We hope these clarifications will further strengthen your already positive evaluation.

---

> > ### Author Rebuttal · Reviewer_CvFZ · 2026-04-02
> >
> > Thanks for the comprehensive rebuttal, and I am more confident in giving this paper a possible score. I therefore raise the confidence score in this paper.

---

> > > ### Author Response · Authors · 2026-04-07
> > >
> > > Thank you very much for the thoughtful follow-up and for your positive assessment. We are especially grateful that, after our rebuttal fully addressed your concerns, you expressed stronger confidence in the paper. Your acknowledgment means a lot to us, and we sincerely appreciate your time, careful reading, and encouraging support.

---

### Official Review · Reviewer_KRN6 · 2026-03-20

**Soundness:** 3
**Presentation:** 3
**Significance:** 2
**Originality:** 2
**Overall Recommendation:** 4
**Confidence:** 4

**Summary:**

This paper focuses on the self-evaluation problem in Large Reasoning Models (LRMs). The authors argue that existing methods mostly rely on output-level signals such as entropy, while overlooking the evolution of internal representations across layers. They hypothesize that correct reasoning is typically accompanied by coherent and directional changes in hidden states throughout the network. To quantify this phenomenon, the paper proposes a white-box framework called GeoR. Specifically, the method extracts the model’s intermediate CoT reasoning process and aggregates the hidden states of reasoning tokens at each layer, treating them as a reasoning trajectory that evolves with network depth. It then uses both the first-order and second-order changes of this trajectory across layers to characterize the reasoning process. These geometric signals are further combined with the average negative log-likelihood of reasoning tokens, which reflects the model’s intrinsic uncertainty and task difficulty, to produce a final self-evaluation score. The computation of this score introduces almost no additional inference overhead.

**Compliance With Llm Reviewing Policy:**

Affirmed.

**Final Justification:**

I think most concerns are resolved and raise the rating accordingly.

**Key Questions For Authors:**

* The method appears to rely on access to an explicit `<think>...</think>` reasoning segment. The paper would be strengthened by including experiments in settings without explicit CoT traces.

* The method’s advantage becomes much less pronounced on larger models (e.g., Qwen-14B). This raises the question of whether the observed gains come entirely from the method itself, or are partly driven by external factors such as model scale. The paper would benefit from additional analysis on the effects of model size, CoT length, problem difficulty, and other related factors.

Overall, I think this paper is not bad. If concerns can be mostly solved, i am open to raise the rating.

**Limitations:**

include clearer limitations, e.g., the method is only applicable in white-box settings.

**Strengths And Weaknesses:**

**Strengths:**

* The method does not require any additional training.
* The experiments are relatively comprehensive.
* The proposed framework is conceptually clear.

**Weaknesses:**

* The evidence that geometric evolution reflects reasoning fidelity is not sufficiently convincing. The authors argue that stronger geometric evolution under conditions such as simpler tasks or correct samples indicates that it captures reasoning fidelity. However, this only provides evidence for sufficiency, not necessity. In other words, the paper does not establish that larger geometric evolution arises from better reasoning rather than from other factors such as response length or reasoning style.

* I do not think the paper sufficiently clarifies its distinction from related work. CoE has already shown that the progressive path of hidden states can be used for output-free self-evaluation; CoRE has explicitly discussed the geometric properties of trajectories; and *Reasoning Models Know When They’re Right* has already demonstrated that hidden states contain correctness signals. Relative to these works, the main contribution of this paper seems to lie more in providing a geometric formulation that improves upon a feature-based view.

* The evaluation metrics are somewhat one-sided. Metrics such as AUROC, AUPR, and FPR@95 mainly measure the ability to distinguish correct from incorrect samples. This still leaves a gap between discrimination ability and a genuinely trustworthy self-confidence estimate. The authors should include more direct calibration-oriented metrics to demonstrate that GeoR can serve as a confidence score that is meaningfully reliable.

---

> ### Author Rebuttal · Authors · 2026-03-30
>
> > **W 1: The evidence that geometric evolution reflects reasoning fidelity is not sufficiently convincing.**
>
> Frankly speaking, establishing a strictly necessary condition for reasoning fidelity is **a common ultimate goal of the research community, and thus remains an open challenge [1,2]**. The whole community including us acknowledge that given the profound architectural complexity of LLMs, the interpretability of their internal computational dynamics is enormously weak. Therefore, no existing protocol can theoretically rule out all potential "unmet conditions" or counter-examples.
>
> Precisely due to the inherent weak interpretability, existing methods—including ours—currently rely on large-scale empirical experiments to validate the effectiveness of CoT evaluation metrics. Importantly, our exhaustive experiments demonstrate that among all existing characterization metrics, GeoR offers the highest accuracy.
>
> Undoudtfully, developing objective and comprehensive metrics for reasoning quality requires continuous research efforts from the community.
>
> [1] Let’s Verify Step by Step. OpenAI
>
> [2] Language Models Don't Always Say What They Think. Anthropic
>
> > **W 2: Distinction from related work.**
>
> While prior works can provide fidelity signals, they exhibit critical limitations: CoE lacks a unified metric, CoRE only identifies specific failure patterns, and probe-based verification requires external training. Translating these preliminary geometric observations into a universal, training-free metric poses a monumental challenge due to the immense scale variations and semantic noise across raw reasoning trajectories.
>
> GeoR overcomes this non-trivial hurdle by rigorously coupling layer-wise geometric progression with difficulty-aware calibration. This delivers a robust confidence score that substantially outperforms baselines and generalizes broadly across diverse model architectures, parameter scales, and multimodal contexts.
>
> > **W 3: Evaluation metrics are one-sided.**
>
> To verify whether GeoR serves as a reliable confidence score rather than just a discriminative metric, we evaluate ECE and Brier Score on MMLU-Pro (R1-LLaMA-8B):
>
>  -|AUROC↑|AUPR↑|FPR95↓|ECE↓|Brier↓
> -|-|-|-|-|-
> Random|50.95|31.25|94.30|.239|.295
> CoE|51.77|33.66|94.12|.185|.257
> **GeoR**|**60.47**|**37.90**|**89.24**|**.096**|**.217**
>
>
> While baselines like CoE struggle with calibration on MMLU-Pro, GeoR achieves **significant discriminative advantages** and a notably lower ECE. This confirms that our core geometric formulation can successfully translate abstract latent dynamics into a highly reliable, well-calibrated confidence metric.
>
> > **Q 1: Evaluation without explicit CoT traces?**
>
> **Yes, GeoR works seamlessly without explicit CoT traces.** We utilized `<think>` tags in our main experiments simply because doing so aligns with the standard conventions of current cutting-edge LLMs. However, special formatting tokens are by no means a prerequisite for our method.
>
> Applying GeoR to standard models without special tags is plug-and-play. For instance, when evaluating Qwen2-7B on GSM8K, we naturally isolated the reasoning trajectory using a standard rule-based parsing function—a default and widely-adopted practice for final answer extraction.
>
> Method|AUROC↑|AUPR↑|FPR95↓
> -|-|-|-
> Random|49.75|68.25|91.30
> CoE|68.31|73.82|84.25
> **GeoR**|**75.47**|**81.20**|**78.24**
>
>
> These results confirm that GeoR works very well with or without explicit system tokens.
>
> > **Q 2: Do the observed gains come entirely from the method itself, or are they partly driven by external factors such as model scale?**
>
> The mentioned external factors such as **model scale, CoT length, and problem difficulty do not directly determine GeoR’s effectiveness**.
>
> We individually tested the GeoR's performance with varying one factor while keeping the others fixed. Specifically, the relevant evidence is provided for **scale (Fig. 5, Tab. 9), difficulty (Tab. 1, Tab. 8, Fig. 4), and CoT length (Tab. 1, Fig. 4)**. Across these settings, GeoR remains consistently competitive, which suggests that its advantage is not tied to any single external condition. These results indicate the gains are **not driven by external factors themselves**. Instead, factors such as model scale and task difficulty may change the **positive/negative sample balance**, while CoT length affects the **complexity of trajectory modeling**. Such factors may change *absolute* scores but do not affect GeoR’s **relative advantage**.
>
> The observation on **Qwen-14B** fits this regime. GeoR does **not degrade with scale**; slightly lower AUROC reflects a harder discrimination setting, as stronger models produce fewer and subtler errors. Similar effects are reported in prior work [1].
>
> [1] *Latent Space Chain-of-Embedding Enables Output-free LLM Self-Evaluation*, ICLR 2025.
>
> We hope these clarifications thoroughly resolve your concerns and positively influence your final evaluation.

---

> > ### Author Rebuttal · Reviewer_KRN6 · 2026-04-01
> >
> > Thanks. I think most concerns are resolved and raise the rating accordingly. However, as R to W1, the paper have clear limitations (that is normal, yet). The paper should include clear section to discuss those limitations.

---

> > > ### Author Response · Authors · 2026-04-07
> > >
> > > Thank you very much for the thoughtful follow-up and for raising the rating. We are glad that the rebuttal resolved most of your concerns and helped clarify the main points of the paper. We also appreciate your suggestion to more clearly discuss the paper’s limitations. We agree that, while such limitations are natural for this problem setting, the paper should state them more explicitly. In the revision, we will add a dedicated limitations section to clearly discuss the scope of GeoR. We sincerely appreciate your positive assessment and constructive feedback.

---

### Decision · Program_Chairs · 2026-04-30

**Decision:**

Accept (regular)

**Comment:**

The paper introduces a white-box self-evaluation framework for large reasoning models called GeoR (Geometry of Reasoning). The framework is based on the insight that layerwise trajectory evolution of hidden states provides a signal for assessing reasoning quality without external supervision. The authors evaluate GeoR comprehensively across 8 models (1.5B to 70B parameters) and 7 benchmarks, demonstrating consistent improvements over baselines.

There was a broad agreement among the reviewers that the paper introduces a novel and intuitive view, that the proposed framework is efficient, and the results are convincing and supported by extensive experiments. Some concerns and points for consideration were raised by the reviewers, including about the implementation of GeoR and the lack of MoE evaluations. However, as indicated by the reviewers, all these were resolved by the authors’ rebuttal.

Overall, this is solid work that tackles an important, timely problem and makes valuable contributions to the community.